# Exotic alleles contribute to heat tolerance in wheat under field conditions

Gemma Molero[1,3,5], Benedict Coombes [2,5], Ryan Joynson[2,4], Francisco Pinto[1], Francisco J. Piñera-Chávez[1], Carolina Rivera-Amado[1], Anthony Hall[2✉] & Matthew P. Reynolds [1✉]

Global warming poses a major threat to food security and necessitates the development of crop varieties that are resilient to future climatic instability. By evaluating 149 spring wheat lines in the field under yield potential and heat stressed conditions, we demonstrate how strategic integration of exotic material significantly increases yield under heat stress compared to elite lines, with no significant yield penalty under favourable conditions. Genetic analyses reveal three exotic-derived genetic loci underlying this heat tolerance which together increase yield by over 50% and reduce canopy temperature by approximately 2 °C. We identified an *Ae. tauschii* introgression underlying the most significant of these associations and extracted the introgressed *Ae. tauschii* genes, revealing candidates for further dissection. Incorporating these exotic alleles into breeding programmes could serve as a pre-emptive strategy to produce high yielding wheat cultivars that are resilient to the effects of future climatic uncertainty.

[1] International Maize and Wheat Improvement Center (CIMMYT), Texcoco 56237, Mexico. [2] The Earlham Institute, Norwich NR4 7UZ, UK. [3] Present address: KWS Momont Recherche, 59246 Mons-en-Pévèle, Hauts-de-France, France. [4] Present address: Limagrain Europe, Clermont-Ferrand, France. [5] These authors contributed equally: Gemma Molero, Benedict Coombes. ✉email: anthony.hall@earlham.ac.uk; m.reynolds@cgiar.org

Wheat is among the most widely cultivated crops in the world with more than 216 million hectares grown annually[1], most of which is produced under temperate conditions[2]. Heat stress is one of the major abiotic stressors that impacts global wheat production, reducing leaf area, crop duration and the efficiency of photosynthesis and respiration[3] as well as reducing floret fertility and individual grain weight[4]. Together, these physiological consequences negatively impact productivity[3] with potential devastating effects. For example, in 2010, Russia saw a 30% reduction in wheat yield during their hottest summer in 130 years[5]. Cases like this could become commonplace as global warming causes temperatures to rise and extreme weather events to become more frequent. Simulations predict that global yields will fall by on average 6% for each 1 °C increase in temperature[6], with some regions reaching 9.1% ± 5.4% per 1 °C rise[7]. Adaptation to future climate scenarios is vital to ensure global food security[8]. Climatic instability, combined with environmental constraints, such as restricted supplies of irrigation water and arable land loss, emphasises the need for breeding strategies that deliver both increased yield potential during favourable cycles and resilience to abiotic stress and environmental constraints.

Such adaptation relies on genetic variation underlying the traits of interest; however, modern elite wheat material typically has limited genetic variation, particularly in the D genome[9], due to historic genetic bottlenecks[10,11] compounded by intensive artificial selection by breeders[12]. A strategy employed by the International Maize and Wheat Improvement Center (CIMMYT) to increase the genetic diversity of wheat pre-breeding material is to incorporate exotic parents in their germplasm via strategic crosses[11,13]. The most common exotic parents used are Mexican and other origin landraces[14] and primary synthetics, which are produced by hybridising tetraploid durum wheat with *Aegilops tauschii*, the ancestral donor of the D genome, to recreate hexaploid bread wheat[15]; these synthetic lines act as a bridge to introduce durum and *Ae. tauschii* variation into modern hexaploid wheat. This approach has been successful in introducing disease resistance as well as drought and heat adaptive traits[16,17]. Landrace and synthetic material have been identified with superior biomass in comparison to elite lines under drought and heat conditions[18,19] and elite lines that include landrace or synthetic material in their background have been developed in recent years for drought, heat, and yield potential conditions[20–22].

Challenges remain for the effective deployment of landrace and synthetic material. Only a small fraction of these vast collections of crop genetic resources have been evaluated for climate resilience traits and potential tradeoffs under favourable conditions have not been assessed. Currently, most of these genetic resources are unused[23] as breeders tend to avoid exotic materials because of large regions of poor recombination and a fear of linkage drag[24]. Furthermore, despite evidence of the contribution of exotic material in wheat improvement, the physiological and genetic bases of heat tolerance in this material remain unclear.

Here, we evaluate a spring wheat panel in the field containing contrasting material controlled for phenology and plant height under heat stress and yield potential conditions. We explore yield and related physiological traits and compare exotic-derived lines with elite lines. We conduct a genome-wide association study to reveal marker trait associations (MTA) with heat tolerance traits and evaluate their impact under favourable conditions. Finally, we identify introgressed *Ae. tauschii* underlying an MTA and employ in silico mapping downstream of the GWAS to narrow down the interval, explore recombination and identify candidate genes.

## Results

**Physiological evaluation of HiBAP I under heat stress.** To estimate the contribution of exotic material to heat tolerance and identify its genetic bases, we evaluated the High Biomass Association Panel I (HiBAP I) for two consecutive years under yield potential and heat stressed irrigated conditions in NW-Mexico (Supplementary Table 1). The HiBAP I panel represents an unprecedented resource of genetic diversity[25]. It contains 149 lines, some of which are elite while others contain exotic material from landraces, synthetics, and wild relatives (Fig. 1a, Supplementary Data 1). All lines have agronomically acceptable backgrounds and a restricted range of phenology and plant height under yield potential conditions[21] which allows traits of interest to be evaluated without confounding effects.

Heat stress was imposed by delayed sowing compared to the check environment (Supplementary Fig. 1) and, across both years of evaluation, this reduced yield by 48.1% and shortened the crop cycle duration by more than 30% (Fig. 1b, Supplementary Table 3). When we analysed the response to heat stress of the lines based on their pedigree, exotic-derived lines exhibited an average of 37.7% higher yield compared to elite lines under heat stressed conditions (Fig. 1c, upper). Biomass, the trait most affected by heat stress, was 39% higher in exotic-derived lines, and other yield components, except for harvest index (HI), were significantly higher in exotic-derived lines than elite lines (Fig. 1c, upper). Under yield potential conditions, exotic-derived lines did not show a yield penalty compared to elite lines, as reported in[21] (Fig. 1c, lower). Exotic lines were on average 5.6 cm and 3.8 cm taller than elite lines under heat stressed and yield potential conditions, respectively. No differences in phenology were observed between the groups in either of the environments. Plant height was not correlated with yield under yield potential conditions ($r = -0.007$, $p > 0.05$), but positive correlations were observed between plant height and yield under heat stressed environments ($r = 0.699$, $p < 0.001$). The better performance of exotic-derived lines was validated using the stress susceptibility index (SSI). This measure is negatively correlated with yield under heat stressed conditions; thus, lower SSI values indicate higher tolerance to a stressful environment. Compared to elite lines, exotic-derived lines had significantly lower SSI values for yield, grains per m² and biomass at physiological maturity, but not for thousand grain weight (Table 1).

Additional physiological traits were measured in the experiments to help understand the physiological basis of the superiority of the exotic-derived lines under heat stressed conditions. Exotic-derived lines had significantly higher normalised difference vegetative index (NDVI), a proxy for biomass, and significantly lower canopy temperature during both vegetative and grain filling stages under heat stressed conditions but not under yield potential conditions (Fig. 2). NDVI measured during vegetative and grain filling stages was positively correlated with yield (Fig. 2) while canopy temperature was negatively correlated with yield at both stages (Fig. 2). These correlations were present under heat stressed conditions but not under yield potential conditions. The correlations were steeper for exotic-derived lines than elite lines for both traits across both phenological stages, suggesting that NDVI and canopy temperature are having a higher impact on yield in exotic-derived lines compared with elite lines. Under heat stressed environments, both NDVI and canopy temperature presented similar correlations with biomass at physiological maturity, grain number, and other yield components (Supplementary Table 5) but no correlation was observed under yield potential. The stress susceptibility index index calculated for yield was negatively correlated with agronomic and physiological traits except for canopy temperature, where positive correlations were observed

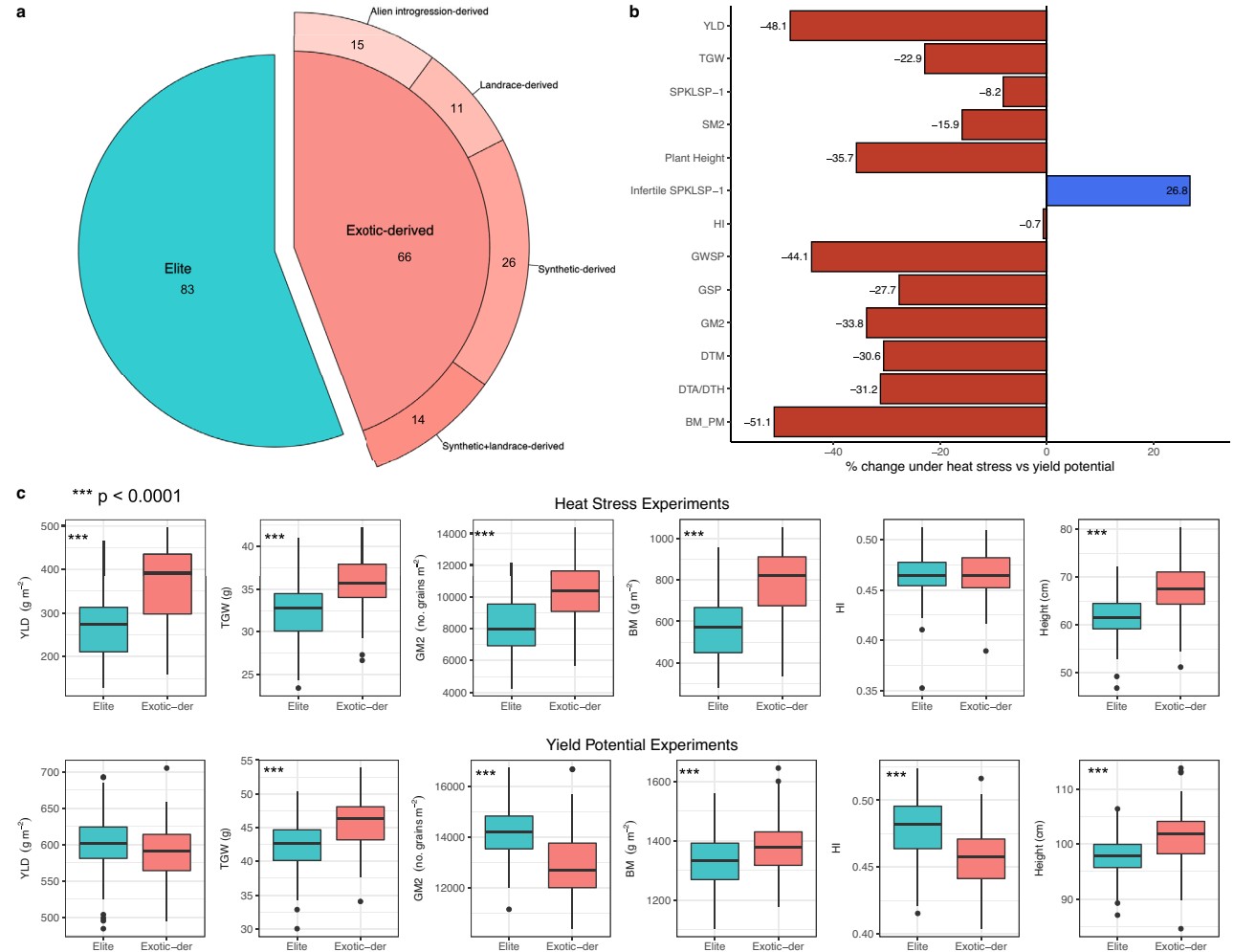

**Fig. 1 Physiological assessment of HiBAP I panel, comparing elite and exotic-derived lines under heat stressed and yield potential conditions.**
**a** Number of lines from each group. **b** Effect of heat stress on yield (YLD), thousand grain weight (TGW), number of spikelets per spike (SPKLSP$^{-1}$), number of spikes per m$^2$ (SM2), plant height (Height), number of infertile spikelets per spike (infertile SPKLSP$^{-1}$), harvest index (HI), grain weight per spike (GWSP), number of grains per spike (GSP), grain number (GM2), days to physiological maturity (DTM), days to anthesis or days to heading (DTA/DTH for yield potential and heat stress experiments respectively), and biomass at physiological maturity (BM_PM), showing the percentage difference compared to yield potential conditions. **c** Comparison of yield (YLD), thousand grain weight (TGW), grain number (GM2), biomass at physiological maturity (BM_PM), harvest index (HI), and Height between elite and exotic-derived lines in HiBAP I measured under both heat stress and yield potential conditions. The boxplots are defined as follows: centre line = median; box limits = upper and lower quartiles, whiskers = 1.5x interquartile range; points = outliers. The significance of the difference between Elite ($n = 83$ biologically independent lines) and exotic-derived ($n = 66$ biologically independent lines) lines for each trait was assessed using two-tailed $t$ tests with no assumption of equal variance. $p$-values below 0.01 were considered significant (*), below 0.001 very significant (**) and below 0.0001 highly significant (***). Means, standard deviations, confidence intervals and $p$-values can be found in Supplementary Table 2.

indicating that more tolerant lines had consistently cooler canopies (Supplementary Table 5).

**Genome-wide association analysis reveals genetic associations under heat stress.** To explore the genetic bases of these exotic-derived heat tolerance traits, marker-trait association analyses were performed using Best Linear Unbiased Estimator (BLUE) means from two or four replicates for each measured trait over two growing seasons. The most relevant MTAs are shown in Supplementary Table 6, and all Manhattan plots are shown in Supplementary Fig. 2. We found 3 pleiotropic markers (Fig. 3a) on chr1B (chr1B-63398861: C), chr2B (chr2B-820002: C) and chr6D (chr6D-6276646: T). These MTAs were associated with all three heat stress indices along with multiple yield traits, including yield and canopy temperature, at both vegetative and grain filling

stages, and were not associated with harvest index or phenology (Fig. 3c, Fig. S2). The favourable allele at each position was the minor allele.

Lines with the favourable C allele on 1B and 2B and the unfavourable A allele on 6D have 24.3% higher yield under heat stress; lines which also have the favourable T allele on 6D have 56.5% higher yield under heat stress compared to lines with the three unfavourable alleles (Fig. 3b). Assuming the three alleles do not interact epistatically, the T allele on 6D can be estimated to increase yield under heat stress by 32.4%. Lines with the favourable allele at all three MTAs show a reduction in canopy temperature of 1.97 °C and 2.37 °C, at vegetative and grain filling stages, respectively, when compared to lines with the unfavourable allele at all three positions (Fig. 3b). Under yield potential conditions, no difference was observed between favourable and unfavourable allele combinations for yield or for canopy temperature (Fig. 3b). The

**Table 1 Stress susceptibility index (SSI) calculated for yield (YLD), thousand grain weight (TGW), grains per m² (GM2) and biomass at physiological maturity (BM_PM) of elite and exotic-derived lines obtained from adjusted means for two years of data in each environment.**

| Trait | $r_p$ (YLD_Heat) | Elite | | Exotic | |
|---|---|---|---|---|---|
| | | n = 83 | | n = 66 | |
| SSI_YLD | −0.976 | 1.16 ± 0.24 | a | 0.79 ± 0.32 | b |
| SSI_TGW | −0.439 | 1.03 ± 0.2 | a | 0.95 ± 0.27 | a |
| SSI_GM2 | −0.951 | 1.26 ± 0.42 | a | 0.60 ± 0.52 | b |
| SSI_BM_PM | −0.946 | 1.13 ± 0.20 | a | 0.85 ± 0.23 | b |

Letters indicate the statistical significance between Elite and Exotic groups. Means followed by different letters are significantly different ($p$-value < 0.01) according to pairwise t tests. $r_p$ corresponds to the phenotypic correlation with the yield obtained under heat environments. Data represents the mean ± S.D. Sample size, n, indicates the number of biologically independent lines in each group.

favourable allele at each of these MTAs is predominantly found in exotic-derived lines with 50/55 (1B), 44/45 (2B) and 33/33 (6D) lines with the favourable allele classified as exotic-derived. 7 lines appear to be heterozygous (A/T) at 6D-6276646. The HiBAP lines are inbred to at least the F9 or F10 generation so, as sequencing data was generated from pooled samples, this observation could be the result of alleles segregating at this locus. However, we observe no significant difference in yield or canopy temperature under heat stress between lines that are heterozygous and lines that are homozygous for this allele (Supplementary Fig. 3). This suggests that these lines are indeed heterozygous for the favourable allele and also suggests that the phenotype may be dominantly inherited.

**Aegilops tauschii introgression underlies 6D MTA**. Due to the better performance of exotic-derived lines under heat stress and exotic-derived lines possessing alleles for heat tolerance, we searched for introgressed material overlapping the MTAs. We detected introgressed material in HiBAP I lines by looking for genomic blocks containing windows with SNPs specific to *Ae. tauschii*, *Th. ponticum* or *S. cereale* and reduced mapping coverage, seen as coverage deviation (mapping coverage compared to the median mapping coverage across the panel) significantly below 1, which indicates breaks in synteny between wheat and the introgressed chromosome segment. Using this approach, we identified introgressed *Ae. tauschii* material at the beginning of 6D in all 33 lines with the T/T genotype and all 7 lines with the A/T genotype at MTA 6D-6276646, where T is the favourable allele. As *Ae. tauschii* is from wheat's primary genome and thus very similar to the D subgenome, not every 1Mbp window is sufficiently lacking in synteny for reads to map poorly and produce significant coverage deviation below 1. This explains why some windows within the introgression have coverage deviation of around 1. However, these windows still have *Ae. tauschii*-specific SNPs and are within a block of 1Mbp windows in which most have significant coverage deviation below 1. Therefore, we can be confident that the introgression includes these windows.

The full-length, unbroken segment is 31.6Mbp in length, as seen in Sokoll (HiBAP_57) (Fig. 4a). The segment size within independent Sokoll Weebil1 crosses show that recombination occurs readily within the segment, breaking it up into variable sizes (Supplementary Fig. 4). By comparing the overlapping segments between lines, we found a 1.80Mbp core introgressed region between 5.05Mbp and 6.85Mbp that is present in all lines with the T/T or A/T genotype at 6D-6276646 and absent in all the lines with the A/A genotype (Fig. 4a). In A/T lines, the introgression itself, in addition to the favourable allele, appears

to be heterozygous, evidenced by intermediate mapping coverage deviation compared to the homozygous lines and by heterozygous SNPs whose alternative alleles are specific to *Ae. tauschii*. Using chromosome and protein alignments, we anchored this 1.80Mbp core region from the wheat RefSeq v1.0 genome to the *Ae. tauschii* reference genome, Aet v4.0[26], and extracted the syntenic 1.49Mbp region between 4.63Mbp and 6.12Mbp. This represents the probable introgressed content of the core introgressed region and likely contains the gene(s) responsible for the MTA (Fig. 4b, c). We found no evidence of introgressed material overlapping the 1B or 2B MTAs.

**Candidate genes for MTAs in 1B, 2B and 6D**. For the 6D MTA, we identified the syntenic region in the *Ae. tauschii* genome and a list of genes that had been introgressed (Fig. 4c). As we are unaware of the *Ae. tauschii* accession that has been introgressed, we also looked at the genes within the same region in four other available chromosome-level *Ae. tauschii* assemblies[27]. Between accessions, this region varies between 1.49Mbp and 1.82Mbp in length and contains between 26 and 33 genes (Supplementary Data 2). These include a MIKC-type MADS-box gene orthologous to *SUPPRESSOR OF OVEREXPRESSION OF CONSTANS 1* (*SOC1*); a mitogen-activated protein kinase (*MAPK*) gene found in two of the *Ae. tauschii* accessions with no orthologue in wheat; and a pair of type-B two-component response regulator receiver proteins, orthologous to type-B *Arabidopsis* response regulators (ARRs) with closest similarity to *ARR-11*. One member of the pair, AET6Gv20025700, appeared to have a myb-binding domain that is missing from the wheat orthologue gene model. However, after manual reannotation, we found that this difference was a misannotation in wheat so likely not causing a functional difference. We also found that both ARR genes were expressed in spike and grain in both *Ae. tauschii* and wheat but not in leaf or root, whereas the other candidate genes were expressed across leaf, root, spike and grain tissues. This might exclude the ARR genes for involvement in the heat tolerant phenotype which is established during the vegetative stage and maintained through grain filling. For the 1B and 2B MTAs, as they were not within an introgression, we submitted the sequence 1Mbp up and downstream of the MTA to Knetminer, a gene discovery tool[28]. Within the 2B interval, we identified *DEHYDRATION-RESPONSIVE ELEMENT-BINDING PROTEIN 1A* (*DREB1A*) and *STEROL GLUCOSYLTRANSFERASE* (*SGT*) as promising candidate genes. The functional evidence of candidacy for each candidate gene is outlined in Supplementary Note 1.

**Discussion**

Exotic parents are routinely used to increase genetic diversity in wheat pre-breeding pipelines and their enhanced performance has been demonstrated under salinity[29], drought[14,19] and heat stress[18,30]. In the present study, exotic-derived lines performed better under heat stress than elite lines with no yield penalty under yield potential conditions. This increased yield under heat stress was associated with a range of factors, including higher biomass throughout the crop cycle, higher grain number and cooler canopy temperature during both vegetative and grain filling stages. Contrary to other studies[31,32] higher pre or post anthesis biomass (NDVI) or lower canopy temperature was not associated with higher yields under favourable conditions. Cooler canopies have been previously associated with higher tolerance to drought and heat irrigated environments[33] and with optimised root distribution in bread wheat[34]. Plants with an optimised root system are able to satisfy the high evaporative demand through elevated transpiration rates under hot irrigated conditions and thus maintain cooler canopies[35]. Higher transpiration rates are associated with increased

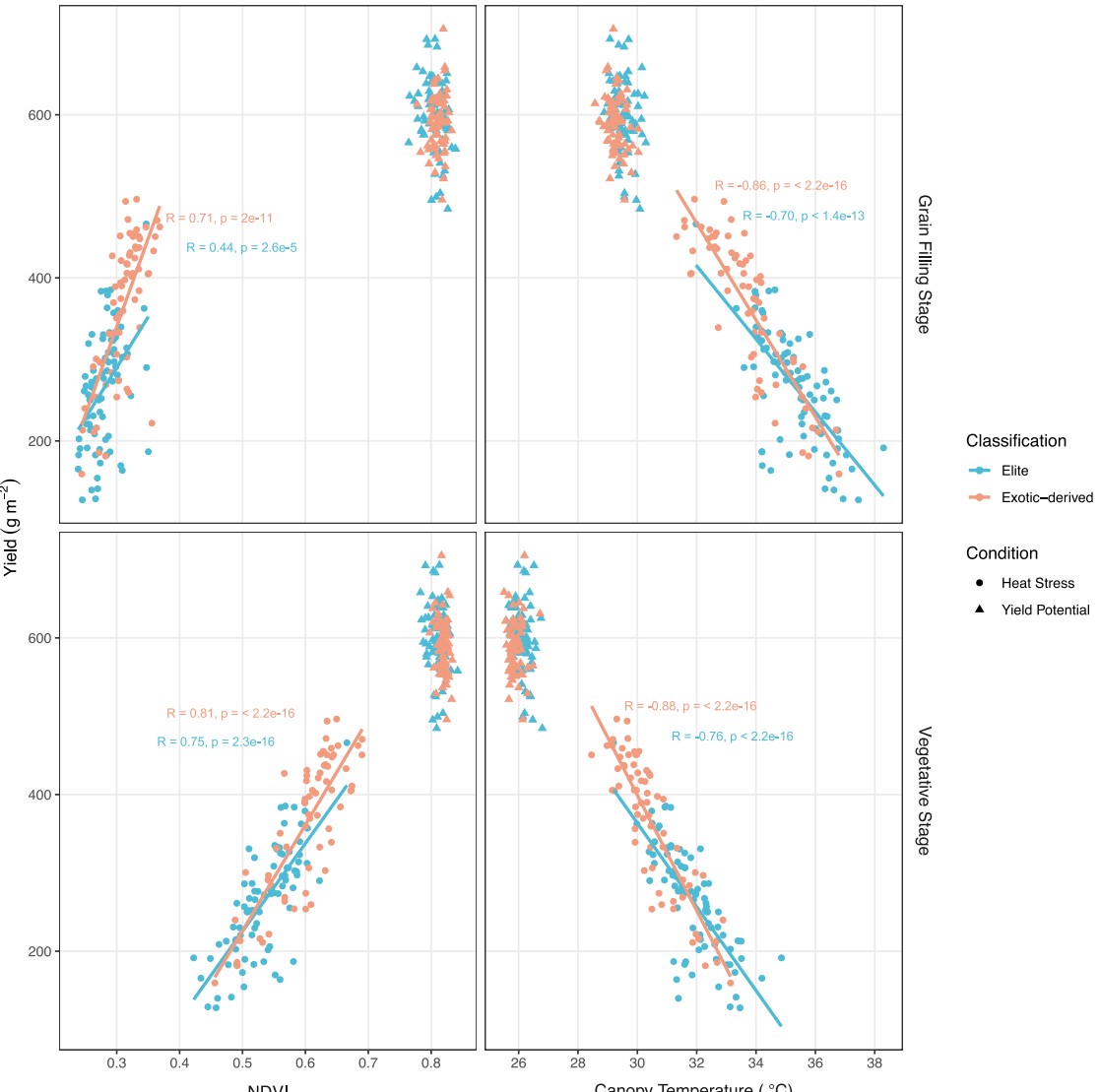

**Fig. 2 Relationship between normalised difference vegetation index (NDVI) and yield and between average canopy temperature and yield at both grain-filling and vegetative stages.** NDVI and canopy temperature were measured with UAVs at pre-heading (vegetative stage) and during grain filling. Regression lines were calculated using Pearson's correlation coefficient between each pair of traits ($n = 83$ and $66$ biologically independent lines for the Elite and exotic-derived groups, respectively) and added for classification/condition combinations with a significant correlation ($p$-value $<= 0.01$). The correlation coefficient, $r$, and the steepness of the line, ranges from $-1$ to $1$, signifying very negatively correlated and very positively correlated, respectively. Pearson's correlation coefficient, confidence intervals and p-values for all comparisons can be found in Supplementary Table 4.

stomatal conductance that, in turn, is associated with higher photosynthesis that can explain the higher biomass observed in exotic-derived lines in comparison with elite lines. However, according to temperature response models in wheat[6], the observed reduction in plant temperature of approximately 2 °C would be unlikely to account alone for the >50% increased yield of exotic lines[6].

Despite variation in plant height being restricted, exotic-derived lines were taller than elite lines in both environments. Plant height and phenology were restricted under yield potential conditions, but the variation under heat stress environments was not initially considered for the panel construction. Interestingly, lines that performed well under heat stress had the lowest difference in plant height between yield potential and heat stress conditions. Taller plants have better light interception and a better light distribution in comparison with shorter plants, and this has been associated with increased photosynthesis[36]. Therefore, plant height may be influencing the better performance of exotic-derivatives. Among all stress indices, the stress susceptibility

index (SSI) is thought to be the most useful index for evaluating tolerant cultivars. Exotic-derived lines had significantly lower SSI than elite lines, adding additional support to the resilience of this exotic material under heat stress.

In the present study, heat stress was achieved by delaying sowing by more than three months. This could have introduced confounding effects as delayed sowing changes not only temperature but also photoperiod. However, the photoperiod effect in this study is considered minimal for several reasons. Firstly, the lines presented in this study were selected using the shuttle breeding technique that characterises CIMMYT's wheat breeding strategy and selects lines relatively insensitive to photoperiod and vernalisation response. This is because one selection site has a long photoperiod and negligible vernalizing cold[37]. Secondly, insensitivity to the photoperiod was confirmed by marker analysis where among the known major adaptation genes, the spring allele at *Vrn-B1* (*Vrn-B1a*) and *Vrn-D1* (*Vrn-D1a*) and the Ppd-insensitive allele at *Ppd-D1* (*Ppd-D1a*) were present in ~90% of

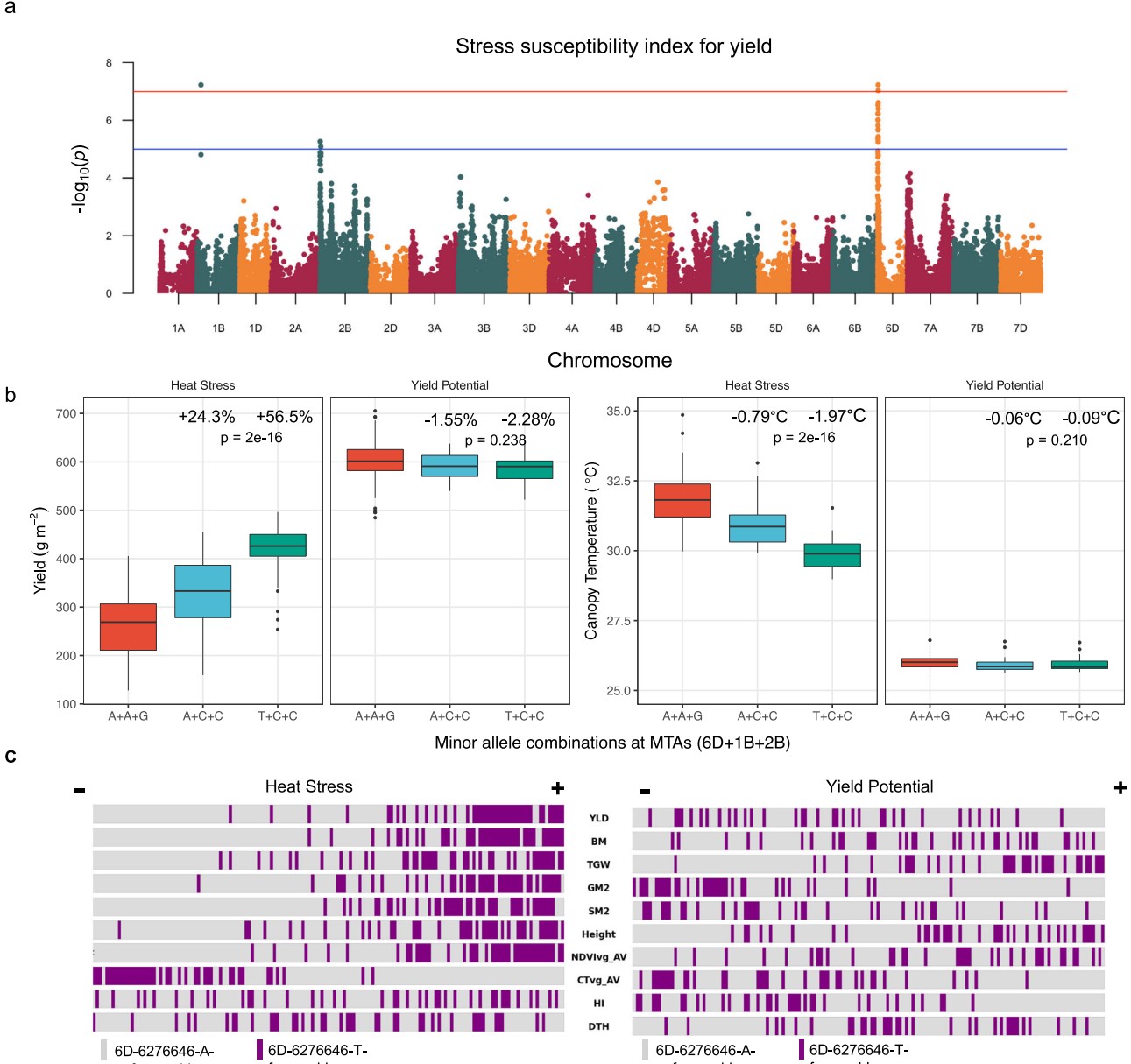

**Fig. 3 Genome-wide association study reveals genetic markers underlying heat tolerance traits. a** Manhattan plot showing marker trait associations for stress susceptibility index (SSI) for yield under heat stress. The horizontal blue line indicates an arbitrary cutoff of $-\log_{10}(p)$ of 5. The horizontal red line indicates the conservative Benjamini–Hochberg cutoff implemented by GAPIT. **b** Specific marker allelic variants effects on yield and on canopy temperature under heat stress and yield potential conditions in chromosomes 6D (chr6D-6276646), 1B (chr1B-63398861), and 2B (chr2B-820002), where the combination of favourable alleles is T+C+C and the combination of unfavourable alleles is A+A+G. The boxplots are defined as follows: center line = median; box limits = upper and lower quartiles, whiskers = 1.5x interquartile range; points=outliers. The percentage change and °C change is calculated compared to lines with the major alleles at all three MTAs. Significance of allele combinations was computed using a one-way ANOVA test ($n$ = 87, 14, and 31 biologically independently lines for A+A+G, A+C+C and T+C+C, respectively). Means, standard deviations and $p$-values from Tukey's honest significance test can be found in Supplementary Table 7. **c** Phenotype distribution under heat stress and yield potential conditions highlighting the rank of 6D minor allele carriers for each phenotype where lines in the panel are ordered from lowest to highest value for each trait.

the HiBAP I panel[38]. Finally, delayed sowing at CIMMYT's Obregon station is routinely used for evaluating heat-tolerant breeding material, and several studies confirm the value of late sowing at this experimental site to develop germplasm adapted to different heat stressed environments worldwide[4,18,39–41].

*De novo* SNP discovery is the process of generating SNP markers from high-throughput next-generation sequencing as opposed to using lower density genotyping arrays. The value of this approach in breeding efforts is starting to be more widely recognised. In conjunction with high throughput phenotyping

methods[42], high density, unbiased markers can be leveraged to discover MTAs or to narrow pre-existing QTL intervals to provide more robust markers for global breeding programs[43,44]. Using these methods, we have identified alleles at three pleiotropic MTAs on chromosomes 1B, 2B and 6D that when stacked increase yield by 56.5% and reduce canopy temperature by 1.97 °C/2.37 °C under heat stress conditions when compared to lines containing the three major alleles at these positions (Fig. 3b). These markers were associated with multiple agronomically important traits under heat stress including yield, grain per

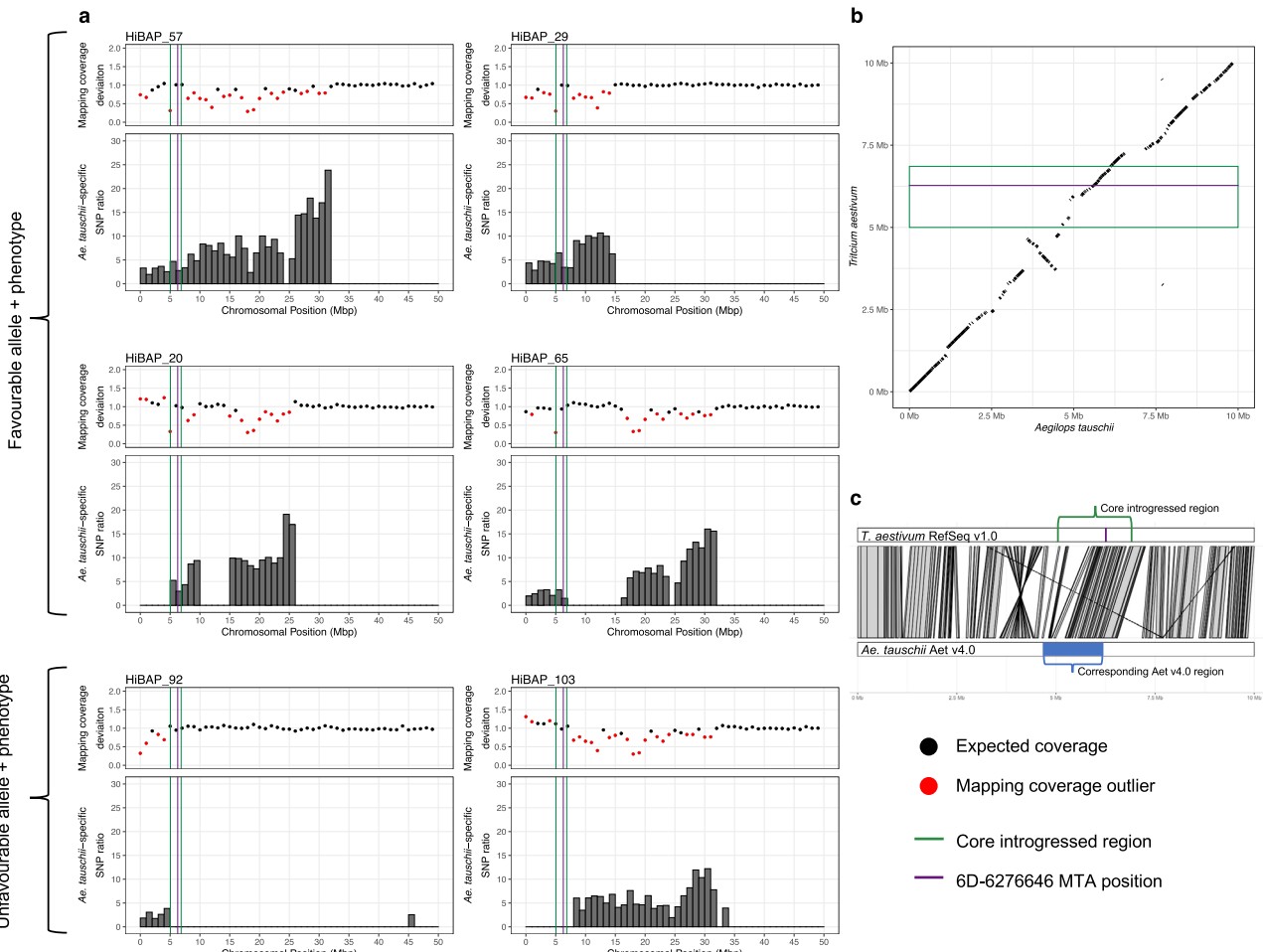

**Fig. 4 *Aegilops tauschii* introgression underlies chr6D-6276646 MTA. a** Visualising *Ae. tauschii* introgressions across the first 50Mbp of chr6D in six HiBAP I lines, four containing the favourable T allele at chr6D-6276646 (HiBAP 57, 29, 48, and 65) and two containing the unfavourable A allele at chr6D-6276646 (HiBAP 92 and 103). Mapping coverage deviation was computed between the HiBAP line and the median of the panel in 1Mbp windows. Red points are statistically significant outliers (*n* = 149 biologically independent lines). *Ae. tauschii*-specific SNP ratio in each 1Mbp window was calculated by dividing the number of homozygous *Ae. tauschii*-specific SNPs in that window by mean number of homozygous *Ae. tauschii*-specific SNPs in that window across the panel and then removing values below 1.45. Green lines mark the borders of the region common to all lines with the favourable T allele, corresponding to a 1.80Mbp region in wheat RefSeq v1.0 and a 1.49Mbp in *Ae. Tauschii* Aet v4.0. The purple line indicates the MTA position. **b** Synteny between 6D:1-10,000,000 in CS RefSeq v1.0[52] and *Ae. tauschii* Aet v4.0[26]. The green box indicates the 1.80Mbp region (1.49Mbp relative to *Ae. tauschii*) common to all lines with the favourable T allele, corresponding to the green region in (**a**). The purple line indicates the MTA position. **c** Alignment of 6D:1-10,000,000 in CS RefSeq v1.0[52] and 6D:1-10,000,000 in *Ae. tauschii* Aet v4.0[26], illustrating how the syntenic region in *Ae. tauschii* was identified and extracted.

square metre, grain filling rate and biomass (Fig. S2). Despite being in apparently disparate regions of the genome, the 1B and 2B favourable alleles always occur together and the 6D favourable allele usually occurs with the 1B and 2B favourable alleles. This suggests that there may be functional linkage between the markers. All three MTAs are predominantly found in exotic-derived lines but are not exclusive to any of the exotic categories as we see them in synthetic, introgression line and landrace derivatives. This brings their origin into question as their most recent pedigree suggests that the favourable alleles may have come from different sources. These MTAs do not overlap with MTAs previously identified for HiBAP I for photosynthetic efficiency[25] or biomass traits[21]. We identified several individuals that appear to be heterozygous for the introgression and for the favourable allele on 6D. As sequencing was conducted on pooled samples of 10 individuals per line, lines that appear heterozygous might instead be segregating for presence/absence of a homozygous introgression. As the phenotype under heat stress appears to be the same

between lines that are homozygous and lines that are heterozygous at this locus, it seems likely that these lines are heterozygous for the introgression and allele. In addition, this also indicates that the heat tolerant phenotype contributed by 6D may be dominantly inherited; however, additional work would be needed to validate this. The zygosity of the allele in these lines can be verified in future work by developing markers and observing how they segregate in subsequent generations.

By utilising mapping coverage information and species-specific SNPs, we identified that the MTA on 6D was within an *Ae. tauschii* introgression. We show that this introgression readily recombines within CIMMYT germplasm by comparing the introgressed segment in different offspring of the same cross. Due to concerns regarding linkage drag and lack of recombination of wild relative introgressions[44], this is promising for the deployment of introgressed segments from the primary genepool into breeding programmes. The recombination enabled us to reduce the size of the interval responsible for the MTA by looking for the

region always present in lines with the favourable genotype. The smallest segment in the panel that contains the MTA is around 5Mbp and can likely be broken down further; the small size and it's telomeric location make it amenable for deployment in breeding programmes.

The longest unbroken segment is present in Sokoll, a commonly used advanced synthetic-derived line. Recombination within the segment takes place in all Sokoll × Weebil1 crosses yet appears unbroken in Sokoll. Therefore, Sokoll may be the donor line for this marker in many of the lines in HiBAP I. This would make sense given its presence in many of the pedigree histories of CIMMYT's synthetic-derived lines (Supplementary Data 1). Some of the HiBAP I lines contain an *Ae. tauschii* segment that contains both the 6D MTA and a resistance gene upstream that underlies an MTA from a recent GWAS in *Ae. tauschii*[45]. If the accession of *Ae. tauschii* in the HiBAP I lines confers the same resistance these lines could be used as donors for both traits.

The 6D MTA uncovered is supported by an MTA for heat tolerance reported in[13,22] nearby on 6D. Singh et al., 2018[13] state that the 6D MTA overlapped with an *Ae. tauschii* introgression, using speculative markers and pedigree-based inference. Here, we confirm this speculation and then demonstrate its ability to recombine and narrow down the introgressed region conferring the heat resilient phenotype through in silico introgression mapping.

Following the identification of the core introgressed region, we extracted the syntenic region from five *Ae. tauschii* chromosome-level assemblies and used these, as opposed to the wheat reference genome, as our source for putative candidate genes underlying the 6D MTA. Extensive literature searches on the introgressed *Ae. tauschii* genes or the wheat genes within the interval (for the 1B and 2B MTAs) uncovered several candidate genes for further dissection. Candidate genes are by their nature speculative but may provide a starting point for follow-up studies aiming to map the causal genes. As the 6D *Ae. tauschii* segment appears to be actively recombining, it should be possible to precisely dissect this region and map the causal gene. Our proposed candidate genes for the 6D MTA differ from the gene proposed by Singh et al.[13]. By using the *Ae. tauschii* genomes rather than relying solely on the wheat reference genome, we have demonstrated that the isoflavone reductase gene Singh et al.[13] proposed is not present in the core introgressed region. This difference and the introgressed candidate gene not found in wheat identified highlight the importance of considering non-reference genomes downstream of a GWAS, particularly when divergent material has been introduced, as the variation underlying the trait of interest might be absent from the reference genome.

These three markers can be deployed into marker-assisted breeding or introgression pipeline programmes to incorporate heat resilience traits into elite cultivars. The fact that no yield penalty was identified under more favourable conditions adds value to their deployment, especially given the negative impact that has been documented in terms of yield stability under increasing temperatures using extensive international data[46]. The donor lines for these markers will be selected using our introgression mapping approach to introduce minimal linkage drag alongside the traits of interest. Efforts to develop KASP markers for the favourable MTA alleles are currently ongoing at CIMMYT. The germplasm is available to the community through IWYP.org request.

## Methods

**Plant material and growth conditions**. The High Biomass Association Mapping Panel HiBAP I consists of 149 spring wheat lines (Supplementary Data 1) and is composed of elite high yielding lines and lines with exotic material in their pedigree history derived from CIMMYT breeding and pre-breeding programs[21]. These exotic lines include primary synthetic derivative lines, containing between 0.5% and 43% donor material[25]; Mexican and other origin landraces derivatives; and Elite lines containing an introgressed segment of *Th. ponticum* on chr7D and/or *S. cereale* on chr1B[25]. The set of Elite lines contain 11 CIMMYT varieties released from 1966 until 2007 and additional lines derived from the systematic screening under yield potential and heat stressed field conditions of CIMMYT breeding and pre-breeding material. This allowed the identification of elite genotypes with favourable expression of traits of interest such as high biomass/RUE at different growth stages including final above ground biomass under both yield potential and heat stressed conditions. In general, pre-breeding material is derived from crosses where one of the parents was selected for expressing low canopy temperature and/ or high yield or biomass under heat stressed environments.

To construct the final panel, a pre-panel consisting of more than 250 lines from different sources were evaluated in the field under favourable conditions; lines with a favourable agronomic background and without extreme height or phenology under yield potential conditions were selected to reduce the confounding effect of extreme phenology or height on the expression of biomass and other traits. HiBAP I was evaluated during 2015/16 and 2016/17 under yield potential (YP16 and YP17) and heat stressed conditions (Ht16 and Ht17). Heat stressed conditions were created with delayed sowing where emergence was registered in March instead of November or December as in a normal growing cycle (Supplementary Table 1, Supplementary Fig. 1).

The field experiments were carried out at IWYP-Hub (International Wheat Yield Partnership Phenotyping Platform) situated at CIMMYT's Experimental Station in Campo experimental Norman E. Borlaug (CENEB) in the Yaqui Valley, near Ciudad Obregon, Sonora, Mexico (27°24' N, 109°56' W, 38 masl) under fully irrigated conditions for both yield potential and heat stressed experiments. The soil type at the experimental station is a coarse sandy clay, mixed montmorillonitic typic caliciorthid. It is low in organic matter and is slightly alkaline (pH 7.7)[47]. Experimental design for all environments was an alpha-lattice. Yield potential experiments consisted of four replicates in raised beds (2 beds per plot each 0.8 m wide) with four (YP16) and two (YP17) rows per bed (0.1 m and 0.24 m between rows respectively) and 4 m long. For heat stressed experiments, two replicates were evaluated for HiBAP I in 2 m × 0.8 m plots with three rows per bed (Supplementary Table 1). Seeding rates were 102 Kg ha$^{-1}$ and 94 Kg ha$^{-1}$ for YP and Ht experiments, respectively. Appropriate weed disease and pest control were implemented to avoid yield limitations. Plots were fertilised with 50 kg N ha$^{-1}$ (urea) and 50 kg P ha$^{-1}$ at soil preparation, 50 kg N ha$^{-1}$ with the first irrigation and another 150 kg N ha$^{-1}$ with the second irrigation. Rainfall, radiation, maximum, minimum and mean temperature by month for all the years of evaluation are presented in Supplementary Fig. 1.

**Agronomic measurements**. Phenology of the plots was recorded during the cycle using the Zadoks growth scale (GS)[48], following the average phenology of the plot (when 50% of the shoots reached a certain developmental stage). The phenological stages recorded were heading for heat experiments (GS55, DTH), anthesis for yield potential experiments (GS65, DTA) and physiological maturity (GS87, DTM) for both experiments. Percentage of grain filling (PGF) was calculated as the number of days between anthesis and physiological maturity divided by DTM.

Plant height was measured as the length of five individual shoots per plot from the soil surface to the tip of the spike excluding the awns. Spike, awn and peduncle length were measured in five shoots per plot before physiological maturity (PM). Fertile (SPKLSP$^{-1}$) and infertile spikelets per spike (InfSPKLSP$^{-1}$) were also counted in five spikes per plot at PM.

At physiological maturity, grain yield and yield components were determined using standard protocols[49]. Samples of 100 (YP16), 50 (YP17) or 30 (Ht16, Ht17) fertile shoots were taken from the harvested area at physiological maturity to estimate yield components. The sample was oven-dried, weighed and threshed to allow calculation of harvest index, biomass at physiological maturity, spikes per square meter, grains per square meter, number of grains per spike and grain weight per spike. Grain yield was determined on a minimum of 3.2 m² to a maximum of 4.8 m² under yield potential experiments and 1.6 m² under heat experiments. In yield potential experiments only, to avoid edge effects arising from border plants receiving more solar radiation, 50 cm of the plot edges were discarded before harvesting. From the harvest of each plot, a subsample of grains was weighed before and after drying (oven-dried to constant weight at 70 °C for 48 h) and the ratio of dry to fresh weight was used to determine dry grain yield and thousand grain weight. Grain number was calculated as (Yield/TGW) × 1000. Biomass at physiological maturity was calculated from yield/HI. Number of spikes per m² was calculated as biomass at physiological maturity /(shoot dry weight/shoot number).

**Unmanned Aerial Vehicle (UAV) for canopy temperature and NDVI estimation**. Aerial measurements data for canopy temperature and NDVI was collected using different aerial platforms. Each year, the logistics and availability determined which platform could be used for measuring the heat trials. A summary of the platforms used, together with the cameras and the achieved resolutions, is presented in Supplementary Table 8. The multispectral and thermal cameras were calibrated onsite by measuring over calibration panels placed on the ground before and after each mission. An exception were the aircraft missions, where a calibration performed at the airfield would not be representative of the trial conditions. The

flights were designed as a regular grid of north-south flightpaths covering the whole trial with images that overlapped 75% in all directions to ensure a good reconstruction of the orthomosaic. The flights were performed under clear sky conditions at solar noon ±2 h.

NDVI and canopy temperature orthomosaics were obtained from the aerial images using the software Pix4D. The orthomosaics were then exported to ArcGIS where a grid of polygons representing each polygon was adjusted on top of the image. To avoid the border effect, the polygons were buffered 0.5 m from the north and south border of the plot. Finally, the pixel values were extracted using the 'raster' package in R. We extracted the value of all the pixels enclosed within each polygon and removed possible outliers and calculated the average per plot.

**Stress tolerance Indices.** To determine the effect of heat stress in the genotypes evaluated across years and panels, Stress susceptibility index (SSI) was calculated using data from yield potential (Yyp) and heat stressed (Yht) experiments as follows (Eq. 1):

$$SSI = \frac{1 - \frac{Yht}{Yyp}}{1 - \frac{\bar{Y}ht}{\bar{Y}yp}} \tag{1}$$

where $\bar{Y}ht$ and $\bar{Y}yp$ are the mean yields of wheat lines evaluated under heat stress and yield potential conditions, respectively[50].

**Statistics and reproducibility.** Data from both panels was analysed by using a mixed model for computing the least square means (LSMEANS) for each line across both years using the program Multi Environment Trial Analysis with R for Windows (METAR[51],). When its effect was significant, DTA/DTH was used as a covariate (fixed effect) except for phenology. Broad sense heritability ($H^2$) was estimated for each trait across both years as follows (Eq. 2):

$$H^2 = \frac{\sigma_g^2}{\sigma_g^2 + \frac{\sigma_{ge}^2}{e} + \frac{\sigma^2}{re}} \tag{2}$$

where r is the number of repetitions, e is the number of environments (years), $\sigma^2$ is the error variance, $\sigma^2 g$ is the genotypic variance and $\sigma^2 ge = G \times Y$ variance. Unpaired $t$ tests for stress index (SSI) were conducted with the means across years to determine if the elite and exotic groups presented statistical differences with $p$-value < 0.001.

**DNA extraction, capture enrichment and genotyping.** All genotyping data was taken from ref. [25]. Flag leaf material from 10 plants per line was collected from field grown plots post anthesis and pooled prior to extraction with a CTAB-based protocol. DNA was extracted using a standard Qiagen DNEasy extraction preparation and quality and quantity assessed using a NanoDrop 2000 (Thermofisher Scientific) and the Quant-iTTM assay kit (Life Technologies). From this DNA, dual indexed Trueseq libraries with an average insert size of 450 bp were produced for each line and enriched using a custom MyBaits 12Mbp (100,000 120 bp RNA probes) enrichment capture synthesised by Arbour Bioscience and using 8x precapture multiplexing. 90,000 of these probes were designed using an island strategy to target regions across the whole genome. A subgenome-collapsed reference was used to design these probe sequences to enable homoeologous regions to be targeted with a single probe, thus expanding the design space. The final 10,000 probes were designed for selected genes, targeting both the gene body and 2Kbp upstream. Post enrichment libraries were sequenced using an S4 flowcell on an Illumina NovaSeq6000 producing 150 bp paired end reads.

Sequencing reads were trimmed and low-quality reads removed. These reads were mapped to the Chinese Spring RefSeq v1.0 wheat reference genome[52] using BWA mem v0.7.13[53]. Samtools v1.4[54] was used to remove unmapped reads, supplementary alignments, improperly paired reads, and reads that didn't map uniquely ($q < 10$). PCR duplicates were removed using Picard's MarkDuplicates[55]. SNPs were called using samtools mpileup and bcftools call[56] with parameter -m. SNPs were filtered using GATK[55] to remove SNPs that were heterozygous, had a quality score <30 or a depth <5. A locus was designated as homozygous reference if no alternative allele was found but 5 or more reads were mapped at that position. To create a set of shared SNPs for use in GWAS, SNPs for all lines were combined and loci with more than 10% missing data and a minor allele frequency (MAF) below 5% were removed. The remaining SNP loci were subjected to imputation using Beagle 5.0[57].

**Genome-wide association study (GWAS).** STRUCTURE v2.3.4[58] was used to genetically infer the population structure of the panel and produce a population structure matrix. An admixture model was selected and run using 30,000 burn-in iterations and 50,000 Markov Chain Monte Carlo (MCMC) model repetitions for assumed subpopulations of 2–10 for 10 randomly selected, seeded iterations for each assumed subpopulation. The delta $k$ method from[59] was applied to all 10 replicates to identify the most likely number of definable subpopulations. This was implemented using the STRUCTURE HARVESTER Python script[60]. Finally, CLUMPP v1.1.2[61] was used with 10 independent STRUCTURE replicates to produce a consensus Q matrix for each assumed subpopulation number. GWAS

analysis was conducted using the MLM model implemented in GAPITv3.0[62]. Principal component analysis eigenvectors 1–10 or membership coefficient matrices for 3-8 assumed subpopulations deduced above by STRUCTURE were used as covariates in the model to mitigate the effects of hidden familial relatedness. The EMMA method[63] was implemented in GAPIT to create a positive semidefinite kinship matrix required by the MLM model. Each MTA flanking interval was deduced by identifying the SNP position furthest upstream and downstream from the highest associated SNP that was above the -log P threshold of 5.

**Identifying regions of divergence.** RefSeq v1.0[52] was split into n genomic windows using bedtools makewindows[64]. Using the alignments produced in ref. [25] and detailed above, the number of reads mapping to each window was computed using hts-nim-tools[65]. To normalise by the sequencing depth of each line, read counts were divided by the number of mapped reads that passed the filters, producing normalised read counts c. Different windows of the genome have variable mapping coverage rates, so to compute coverage deviation we must compare each window to the same window in the other lines in the collection. Median normalised read counts, m, were produced, containing the median for each genomic window across the 149 lines. Mapping coverage deviation, d, was then defined for each line as follows (Eq. 3):

$$d_i = \frac{c_i}{m_i \cdot \varepsilon} \tag{3}$$

for window $i \in \{1, 2, \ldots, n\}$, where $\varepsilon$ is the median $d$ value across the genome for the line. Statistically significant $d$ values were calculated using the scores function from the R package 'outliers' with median absolute deviation (MAD) and probability of 0.99. This method was based on ref. [66].

**Producing species-specific SNPs.** Paired-end whole-genome sequencing data for the *Ae. tauschii* reference accession AL8/78[26] and 5 additional accessions that represent 5 different clades[27], 4 *Secale cereale* accessions[67], *Secale. vavilovii*[67], *Thinopyrum ponticum*[68], and *T. aestivum* cultivars Weebil[68], Norin61[68] and Pavon76[69] were mapped to RefSeq v1.0[52], filtered and SNP called as described for the genotyping above and in[25]. Homozygous SNPs were retained if they had between 10 and 60 reads supporting the alternative allele and an allele frequency > = 0.8. Heterozygous SNPs were retained if they had between 10 and 60 reads mapped and were biallelic with each allele having > = 5 reads in support and an allele frequency > = 0.3. SNPs from one relative species not shared with any of the other species or wheat cultivars were retained as species-specific SNPs. These species-specific SNPs were assigned to HiBAP SNPs if they matched in position and allele. Species-specific SNP ratios were calculated by dividing the number of SNPs in each window matched to a species-specific SNP by the mean number of SNPs matched to that species in that window across HiBAP I. SNP ratio scores below 1.45 were removed to keep enriched scores only.

**Synteny between Ae. tauschii and T. aestivum.** The first 10 Mb of 6D from CS and *Ae. tauschii* Aet v4.0[26] were aligned using Minimap2[70] with parameters -x asm10. Alignments <2.5 Kb in length or with mapping quality <40 were discarded. The dot plot was produced using pafr R package[71]. Proteins encoded by genes in the first 10Mbp of 6D in *Ae. tauschii* and CS were aligned using BLASTp[72]. Protein alignments and minimap2 alignments were used to anchor either side of the region commonly introgressed in all lines with the 6D T genotype to anchor the region from CS to *Ae. tauschii*. The *Ae. tauschii* genes and their proteins within this segment are considered as candidate genes. BLASTp[72] was used to compare these proteins to wheat proteins. Protein domains were identified using HMMER hmmscan[73] via ebi using Pfam, TIGRFAM, Gene3D, Superfamily, PIRSF, and TreeFam databases.

**Extracting corresponding region and genes from Ae. tauschii genomes.** Proteins encoded by genes in the first 10Mbp of 6D in *Ae. tauschii* and CS were aligned using BLASTp[72]. Protein alignments and minimap2 alignments were used to anchor either side of the region commonly introgressed in all lines with the 6D T allele to anchor the region from CS to *Ae. tauschii*. The sequence extracted from the *Ae. tauschii* reference genome was aligned to the other 4 chromosome-level assemblies using minimap2[70] with parameters -x asm5. Alignments below length 5000 or quality of 40 were removed. The coordinates of each orthologous region were determined manually and the genes within these coordinates extracted from the respective gff files. The *Ae. tauschii* genes and their proteins within these segments were considered as candidate genes for functional exploration. BLASTp[72] was used to compare these proteins to wheat proteins. Protein domains were identified using HMMER hmmscan[73] via ebi using Pfam, TIGRFAM, Gene3D, Superfamily, PIRSF, and TreeFam databases. Novelty of genes was determined by aligning the extracted protein sequence to each genome using tblastn[72].

**Exploring functionality of candidate genes.** The genes in each identified interval (except for those in the 6D interval) were submitted to Knetminer[28]. The knowledge networks created for each gene were then studied to identify links to the trait from which each MTA was deduced including their biochemical function and orthologous genes being linked in other organisms such as Rice and *Arabidopsis*

thaliana. For the Ae. tauschii genes introgressed into the 6D interval, we conducted extensive literature searches to identify genes with links to heat stress response based on functional studies of related genes.

**Reannotating candidate gene and assessing tissue-specific expression**. To test whether the missing myb-binding domain in the TraesCS6D02G014900 annotation was real or an artefact, we manually reannotated the gene. We identified the exon containing the MYB-binding domain in the wheat orthologue by aligning the coding sequence from the tauschii orthologue to Chinese Spring RefSeq v1.0[52] using tblastn[72]. We mapped Chinese Spring RNAseq data from leaf, root and shoot to RefSeq v1.0[52] using HISAT2[74] and assembled transcripts using cufflinks[75]. We visually inspected the coding sequencing and RNA-Seq alignments using IGV[76], which showed that the MYB-binding domain exon is present and expressed in wheat. To check whether the protein has a premature stop codon, we extracted the coding sequence from the assembled transcript and checked for the presence of a complete open reading frame with no stop codons using EMBOSS getorf[77]. Finally, we checked the presence of intact domains with HMMER hmmscan[73] via ebi using Pfam, TIGRFAM, Gene3D, Superfamily, PIRSF, and TreeFam databases. To explore qualitative expression of candidate genes, we mapped Ae. tauschii RNAseq data from leaf, root, seedling and developing grain 10dd (PRJEB23317) to Aet v4.0[26] as above and abundances were counted using StringTie[78], taking the mean transcripts per million (TPM) across the replicates. Qualitative expression of the CS orthologues was explored using Wheat Expression Browser[79] and the previously leaf, root and shoot RNAseq data mapped above.

**Reporting summary**. Further information on research design is available in the Nature Research Reporting Summary linked to this article.

## Data availability

Publicly available sequencing data used in this study is available at the European Nucleotide Archive (ENA): HiBAP I enrichment capture sequencing data - PRJEB38874; *Th. ponticum*—SRR13484812; *S. vavilovii*: ERR505040, ERR505041, ERR505042; *S. cereale* accession Lo90: ERR504990, ERR504991, ERR504992; *S. cereale* accession Lo176: ERR505005, ERR505006, ERR505007; *S. cereale* accession Lo282: ERR505015, ERR505016, ERR505017; *S. cereale* accession Lo351: ERR505035, ERR505036, ERR505037; *Ae. Tauschii* accession XJ65: SRR13961980; Y173: SRR13962062; SX60: SRR13962012; AY29: SRR13961834; KU2832: SRR13961928; Y215: SRR13962048; Weebil1: PRJEB35709; Norin61: PRJNA492239; Pavon76: https://opendata.earlham.ac.uk/wheat/under_license/toronto/Hall_2021-10-08_wheatxmuticum/PIP-2495/200812_A00478_0126_AHN5W3DRXX/A10948_1_1/; *Ae. tauschii* RNAseq data: PRJEB23317; *T. aestivum* cv. Chinese Spring RNAseq data: Root - SRP133837; SRR6799264; SRR6799265; Leaf - SRR6799258; SRR6799259; SRR6799260; Spike - SRR6802608; SRR6802609; SRR6802610; SRR6802611.

VCF and hapmap genotype files for HiBAP I are available at: https://opendata.earlham.ac.uk/wheat/under_license/toronto/Hall_2022-04-08_HiBAP_genotyping/

Phenotypic data presented in this paper for the HIBAP I panel evaluated under yield potential and heat stressed environments can be found in the Dataverse CIMMYT Research Data Repository at https://data.cimmyt.org/dataset.xhtml?persistentId=hdl:11529/10548643[80].

The source data used to generate the main figures can be found on zenodo at https://zenodo.org/record/7333888#.Y3dmbILP1O6[81], the GitHub repository: https://github.com/benedictcoombes/Exotic_alleles_contribute_to_heat_tolerance_in_wheat_under_field_conditions and in Supplementary Data 3.

## Code availability

The code needed to reproduce the main figures can be found on Zenodo at https://zenodo.org/record/7333888#.Y3dmbILP1O6[81] and at the github repository: https://github.com/benedictcoombes/Exotic_alleles_contribute_to_heat_tolerance_in_wheat_under_field_conditions.

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

## Acknowledgements

This research was supported by the International Wheat Yield Partnership (IWYP) and by the Sustainable Modernization of Traditional Agriculture (MasAgro) an initiative from the Secretariat of Agriculture and Rural Development (SADER) and CIMMYT. Foundation for Food and Agriculture Research under the Grant ID: DFs-19-0000000013. A.H. was supported by BBSRC Core Strategic Programme Grant (Genomes to Food Security) BB/CSP1720/1; A.H. and R.J. was supported by the BBSRC Designing Future Wheat grant BB/P016855/1, BBS/E/T/000PR9783 (DFW WP4 Data Access and Analysis). A.H. and R.J. were also supported by BBSRC/IWYP BB/N020871/1. B.C. was supported by the BBSRC funded Norwich Research Park Biosciences Doctoral Training Partnership grant BB/M011216/1.

## Author contributions

A.H., G.M. and MPR conceived of the idea and designed the experiment. G.M., F.P., F.J.P.C. and C.R.A. collected the field data. G.M. analysed the physiological data. R.J. conducted the genome-wide association study and Knetminer searches. B.C. conducted introgression analysis and introgressed candidate gene searches. B.C., G.M. and R.J. wrote the manuscript. A.H. and M.P.R. were responsible for funding and supervision. All authors reviewed the manuscript.

## Competing interests

The authors declare no competing interests.
