## [Peer Review File · Communications Biology]

Reviewers' comments:

Reviewer #1 (Remarks to the Author):

Review of : Exotic alleles contribute to heat tolerance in wheat under field conditions

Brief Summary: A 149 entry High Biomass Association mapping panel that included derivatives of progenitor species and other wild wheats was evaluated under heat stress and non stress conditions in Obregon, Mexico for 2 successive seasons, harvest years 2016 and 2017. GWAS of the mapping panel indicated 3 exotic loci that increased heat tolerance and yield by 50% and reduced canopy temperature. The 3 putative loci were not associated with harvest index or phenological traits. The locus on chromosome 6 D was associated with an *Aegilops tauschii* introgression. The two other loci located on 1B and 2B were not evidently associated with an introgression. It was noted however that the minor alleles at these loci were always associated together and usually with the minor allele from the locus on 6D.

Significance: This work is significant because it represents one of the first steps toward accessing the wild wheat germplasm in breeding for climate resilience. While the progenitor species have been widely used as sources of pest resistance genes, there have been few such efforts in the area of climate resilience genes. All in all, a very solid, long needed paper.

Comments:

- 1) I would like to hear more about the mapping panel. There must have been preliminary work in the context of breeding for heat tolerance and other resilience traits that gave rise to this panel. Please provide more information.
- 2) Was a Bonferroni or some other correction factor used in assessing the significance of SNPs that appeared in the GWAS? Please comment.
- 3) I am confused about height variation in the panel – the authors say that the variation for height and heading date was constrained in the panel – I understand the logic here – but yet there was a significant correlation between yield and height under heat stress. Please address this.
- 4) The authors state: "Under heat stressed environments, both NDVI and CT presented similar correlations with BM_PM, GM2 and other yield components (Table S4) but no correlation was observed under yield potential." Not sure I understand this. In most studies with which I am familiar, under non stress conditions there is a strong positive correlation between NDVI and yield and related traits. Please clarify.
- 5) The authors state: "These three markers can be deployed into marker-assisted breeding or introgression pipeline programmes to incorporate heat resilience traits into elite cultivars." Are there plans or is there work underway to develop KASP markers for these 3 promising genes?

Reviewer #2 (Remarks to the Author):

Molero et al explore temperature sensitivity in a panel of 149 wheat genotypes by comparative phenotyping in optimal (yield potential) and temperature stressed environments in multiple reps over two different growth years. The temperature stress was induced by late planting (March compared to December in Mexico). The same population has already been used in at least a couple of previous publications (Molero et al *Plant Biotechnol. J.* 17, 1276–1288 (2019), Joynson et al *Plant Biotechnol. J.* 19, 1537–1552 (2021) and detail on the genetic characterisation of the population described in these. The population effectively consists of two sub-populations – the majority (83) being elite hexaploid wheat cultivars and the minority a collection of lines containing exotic genetic material (66) from Alien introgressions, landraces or wheat synthetics. The manuscript initially describes phenotypic performance differences between the two sets of material under stress/non stress conditions then

uses the information to conduct GWAS and more detailed exploration of three strong genetic associations seen across the panel, paying particular attention to a region on chromosome 6D derived from *Ae. Tauschii*. The data convincingly show that these three regions of association have pleiotropic effects on a number of phenotypic traits (by co-location of the genetic associations for multiple phenotypes) under the experimental conditions applied, which is the key message. Unless I am mistaken, they were only able to show that one of these – that on 6D – was derived from an exotic donor (though I did have some issues with their use of the terms minor and major alleles which could be clarified). Exploring the gene content of the region around the 6D introgression introduced a lot of speculation about candidate genes that could be responsible for the observed temperature tolerance. This component could be played down in my opinion. Overall I thought it was very well written and would have been generally understandable if the reader had followed the earlier publications on the same population by the same group – but sometimes I thought it would be difficult to grasp if you hadn't. Some clarity and explanation could therefore be incorporated throughout to allow the first-time reader to understand this paper without having to read the other related papers first. Heat tolerance in the major cereals is of course an increasingly important topic and the data presented, story told and germplasm described in this manuscript is both timely and advance the field.

An important issue that I perhaps missed is how the germplasm will be made available to the science/breeding community.

The statistical analysis of the data look appropriate to me.

Overall, I raise the following points for consideration:

1. As pointed out by the authors the experimental setup induced changes to a whole number of physiological and developmental processes – reducing the crop cycle by 30% and no doubt inducing responses to daylength, humidity, light etc – i.e. not just temperature. It would have been interesting to see how flowering times varied across the two experimental conditions as this may give an indication the impact of factors other than temperature. The authors should comment upon the fact that there may be certain confounding issues in their data.
2. Despite being provided with the cross name, selection history and pedigree in the supplementary table, it may have been good to explain a bit more in general terms what the different classes of exotic material were: e.g. do the synthetics all contain a complete D-genome from a diploid wild relative? Do the alien introgressions have single chromosomal segments introduced? Are the elites all contemporary wheat cultivars? I suggest inserting a sentence/section to provide an overview of the genetic composition of the exotics.
3. I had some difficulty understanding Figure 4a and the issues associated with genotyping the lines by TENSEQ. (e.g. Why do all the core introgressed regions – esp the key MTA - have 'expected coverage?'). I appreciate I may simply not understand what's been done – but if that's the case I have to assume that many others will be in the same boat. Of course, it would have been a lot easier had the original parents been recorded, maintained and genotyped alongside the experimental lines and not having these could be considered a significant flaw. I had to go back and read the earlier papers to try to grasp what had been done and why. The point is I wouldn't expect a reader to have to do this – so despite the genotypic information having been described previously - I would recommend the authors take some time to carefully craft a summary description of what was done – and why it had needed to be done that way - in terms that are understandable by the first time reader.
4. It wasn't clear to me if the HiBAP lines that were declared as heterozygous on 6D were actually heterozygous. Could they have been mixed samples due to sampling leaves from 10 different plants? Can you distinguish between a heterozygote and a paralogous duplication? Could they be segmental additions located elsewhere in the genome? Why would they be maintained as hets (chance?). Was segregation analysis conducted to test for true heterozygosity? A little discussion around the hets may be valuable.
5. I appreciate the desire to try to link regions of genetic association with candidate genes but personally I would play this down a bit – especially for the 1B and 2B associations – as it is all very speculative (was this 2B association the same as detected in Joynton et al 2021 for total chlorophyll content). The good news is that the 6B region is apparently recombinationally active so narrowing the interval genetically should be possible leading to delineation of strong and plausible candidates.

Minor comments:

Linking to the results (MTAs) of work done previously on the same population could be more apparent.

Figure 1: Need a labels in the figure legend

P7 - MTA – expand to Marker Trait Association on first use

P8 – explain how you define minor and major alleles on first use – e.g. were the minor alleles always from the exotic lines? Were the 1B and 2B positive alleles from the exotic lines (I missed this)

L178 word missing (introgression??)

please clarify how does the 1.49Mbp Ae. Tauschii segment relate to the 5.05 Mbp region that you previously defined by using the borders of overlapping introgressions

L189 there is a T missing after Chr6D – 6276646

L205 – delete 'from'

L211 – delete 11

L245 – first mention of de novo SNP discovery need to introduce – even if repeating what has been said in other publications

L 390-391 – replace 'confirm details'

L 480 insert AND

Finally: Why were the genes in the 6D region not submitted to Knetminer

We would like to thank the reviewers for their constructive feedback on this work and we hope we have addressed all the comments. We feel that the manuscript has improved as a result.

Below are detailed responses to all comments from the reviewers. Comments are in black, and our responses are in red. Where appropriate, we have referred to the line numbers in the revised manuscript where the changes have taken place. In the revised manuscript, changes are indicated by red text.

Reviewer #1 (Remarks to the Author):

Review of : Exotic alleles contribute to heat tolerance in wheat under field conditions

Brief Summary: A 149 entry High Biomass Association mapping panel that included derivatives of progenitor species and other wild wheats was evaluated under heat stress and non stress conditions in Obregon, Mexico for 2 successive seasons, harvest years 2016 and 2017. GWAS of the mapping panel indicated 3 exotic loci that increased heat tolerance and yield by 50% and reduced canopy temperature. The 3 putative loci were not associate with harvest index or phenological traits. The locus on chromosome 6D was associated with an *Aegilops tauschii* introgression. The two other loci located on 1B and 2B were not evidently associated with an introgression. It was noted however that the minor alleles at these loci were always associated together and usually with the minor allele from the locus on 6D.

Significance: This work is significant because it represents one of the first steps toward accessing the wild wheat germplasm in breeding for climate resilience. While the progenitor species have been widely used as sources of pest resistance genes, there have been few such efforts in the area of climate resilience genes. All in all, a very solid, long needed paper.

Comment 1: *I would like to hear more about the mapping panel. There must have been preliminary work in the context of breeding for heat tolerance and other resilience traits that gave rise to this panel. Please provide more information.*

The following information on the development of the panel in relation to heat tolerance has been added in the materials and methods section (lines 366-373 in revised manuscript):

“The set of Elite lines contain 11 CIMMYT varieties released from 1966 until 2007 and additional lines derived from the systematic screening under yield potential and heat stressed field conditions of CIMMYT breeding and pre-breeding material. This allowed the identification of elite genotypes with favourable expression of traits of interest such as high biomass/RUE at different growth stages including final above ground biomass under both yield potential and heat stressed conditions. In general, pre-breeding material is derived from crosses where one of the parents was selected for expressing low canopy temperature and/or high yield or biomass under heat stressed environments.”

Comment 2: Was a Bonferroni or some other correction factor used in assessing the significance of SNPs that appeared in the GWAS? Please comment.

GAPIT implements Benjamini-Hochberg testing which is quite strict to limit the false discovery rate. The threshold for significance using this multiple testing correction is illustrated by the red line on the Manhattan plots. The blue line on the Manhattan plots indicates a more arbitrary threshold that is often used for genome-wide association studies and is based on the number of SNPs. For clarity, we have added the following sentence in the legend of figure 3 and figure S2:

“The horizontal blue line indicates an arbitrary cutoff of $-\log_{10}(p)$ of 5. The horizontal red line indicates the conservative Benjamini-Hochberg cutoff implemented by GAPIT”

Comment 3: I am confused about height variation in the panel – the authors say that the variation for height and heading date was constrained in the panel – I understand the logic here – but yet there was a significant correlation between yield and height under heat stress. Please address this.

The reviewer is completely right. Plant height and phenology were only constrained under yield potential conditions but not under heat stressed environments. To construct the panel, a pre-panel consisting of more than 250 lines from different sources were put together in the field under favourable conditions and there, extremes in plant height and phenology were discarded to end up with the final panel. Therefore, even though plant height and phenology were restricted under yield potential conditions, the variation under heat stress environments was larger. Interestingly, we have observed in other experiments that the plant height loss between yield potential and heat stressed environments has a genotypic effect and it is associated with yield under heat stress. The plants that are taller under heat stress tend to produce higher yield.

We have added in lines 267-270 that plant height and phenology were only restricted under yield potential conditions:

“Plant height and phenology were restricted under yield potential conditions, but the variation under heat stress environments was not initially considered for the panel construction. Interestingly, lines that performed well under heat stress had the lowest difference in plant height between yield potential and heat stress conditions.”

The following was also added into the material and methods section (lines 374-377 in revised manuscript):

“To construct the final panel, a pre-panel consisting of more than 250 lines from different sources were evaluated in the field under favourable conditions; lines with a favourable agronomic background and without extreme height or phenology under yield potential conditions were selected to reduce the confounding effect of extreme phenology or height on the expression of biomass and other traits.”

Comment 4: The authors state: “Under heat stressed environments, both NDVI and CT presented similar correlations with BM_PM, GM2 and other yield components (Table S4) but no correlation was observed under yield potential. “ Not sure I understand this. In most studies with which I am familiar, under non stress conditions there is a strong positive correlation between NDVI and yield and related traits. Please clarify.

The reviewer is right that previous studies in CIMMYT, Mexico found a correlation between NDVI and yield under favourable conditions (Tattaris et al., 2016 FPS: 7, 1131) and between NDVI and CT (Rutkoski et al., 2016, G3: 6:2799) but these correlations were always smaller compared with heat stressed environments. In this study, no correlation was observed between NDVI and yield or other yield components for NDVI averaged before and after anthesis but also no correlation was observed in the individual dates of measurements in any of the years under yield potential conditions, contrary with what was observed under heat stressed conditions. On top of the NDVI and CT data captured with the drone we also had ground-based measurements for both traits before and after anthesis. No correlation between ground NDVI or CT with grain yield was identified either in those measurements. This can be associated with the population studied or the specific environmental conditions of the years of the study. We have added a sentence in the discussion to address this (lines 255-257 in revised manuscript):

“Contrary to other studies (32, 33) higher pre or post anthesis biomass (NDVI) or lower canopy temperature was not associated with higher yields under favourable conditions.”

Comment 5: The authors state: “These three markers can be deployed into marker-assisted breeding or introgression pipeline programmes to incorporate heat resilience traits into elite cultivars.” Are there plans or is there work underway to develop KASP markers for these 3 promising genes?

Efforts to develop KASP markers using the MTA peak marker sequence are ongoing at CIMMYT for 1B, 2B and 6D. In addition, a joint proposal with the objective to design diagnostic markers has been recently submitted where, in addition, the lines carrying the favourable alleles will be tested under drought conditions. A sentence addressing this has been added at the end of the discussion (lines 356-357 in revised manuscript).

“Efforts to develop KASP markers for the favourable MTA alleles are currently ongoing at CIMMYT.”

Reviewer #2 (Remarks to the Author):

Molero et al explore temperature sensitivity in a panel of 149 wheat genotypes by comparative phenotyping in optimal (yield potential) and temperature stressed environments in multiple reps over two different growth years. The temperature stress was induced by late planting (March compared to December in Mexico). The same population has already been used in at least a couple of previous publications (Molero et al *Plant Biotechnol. J.* 17, 1276–1288 (2019), Joynson et al *Plant Biotechnol. J.* 19, 1537–1552 (2021) and detail on the genetic characterisation of the population described in these. The population effectively consists of two sub-populations – the majority (83) being elite hexaploid wheat cultivars and the minority a collection of lines containing exotic genetic material (66) from Alien introgressions, landraces or wheat synthetics. The manuscript initially describes phenotypic performance differences between the two sets of material under stress/non stress conditions then uses the information to conduct GWAS and more detailed exploration of three strong genetic associations seen across the panel, paying particular attention to a region on chromosome 6D derived from *Ae. Tauschii*. The data convincingly show that these three regions of association have pleiotropic effects on a number of phenotypic traits (by co-location of the genetic associations for multiple phenotypes) under the experimental conditions applied, which is the key message. Unless I am mistaken, they were only able to show that one of these – that on 6D – was derived from an exotic donor (though I did have some issues with their use of the terms minor and major alleles which could be clarified). Exploring the gene content of the region around the 6D introgression introduced a lot of speculation about candidate genes that could be responsible for the observed temperature tolerance. This component could be played down in my opinion. Overall I thought it was very well written and would have been generally understandable if the reader had followed the earlier publications on the same population by the same group – but sometimes I thought it would be difficult to grasp if you hadn't. Some clarity and explanation could therefore be incorporated throughout to allow the first-time reader to understand this paper without having to read the other related papers first. Heat tolerance in the major cereals is of course an increasingly important

topic and the data presented, story told and germplasm described in this manuscript is both timely and advance the field.

An important issue that I perhaps missed is how the germplasm will be made available to the science/breeding community.

The following sentence has been added at the end of the discussion (lines 357-358 in revised manuscript):

“The germplasm is available to the community through IWYP.org request.”

The statistical analysis of the data look appropriate to me.

Overall, I raise the following points for consideration:

Comment 1: As pointed out by the authors the experimental setup induced changes to a whole number of physiological and developmental processes – reducing the crop cycle by 30% and no doubt inducing responses to daylength, humidity, light etc – i.e. not just temperature. It would have been interesting to see how flowering times varied across the two experimental conditions as tis may give an indication the impact of factors other than temperature. The authors should comment upon the fact that there may be certain confounding issues in their data.

In Figure 1 b we can see how days to heading were reduced by 31.2% under heat stressed compared with favourable environments.

We agree that this point has not been addressed in the manuscript and a whole paragraph has been added in the discussion section (lines 276-287 in revised manuscript):

“In the present study, heat stress was achieved by delaying sowing by more than three months. This could have introduced confounding effects as delayed sowing changes not only temperature but also photoperiod. However, the photoperiod effect in this study is considered minimal for several reasons. Firstly, the lines presented in this study were selected using the “shuttle breeding” technique that characterizes CIMMYT’s wheat breeding strategy and selects lines relatively insensitive to photoperiod and vernalization response. This is because one selection site has a long photoperiod and negligible vernalizing cold (38). Secondly, insensitivity to the photoperiod was confirmed by marker analysis where among the known major adaptation genes, the spring allele at *Vrn-B1* (*Vrn-B1a*) and *Vrn-D1* (*Vrn-D1a*) and the Ppd-insensitive allele at *Ppd-D1* (*Ppd-D1a*) were present in ~90% of the HiBAP I panel (39). Finally, delayed sowing at CIMMYT’s Obregon station is routinely used for evaluating heat-tolerant breeding material, and

several studies confirm the value of late sowing at this experimental site to develop germplasm adapted to different heat stressed environments worldwide (4, 18, 40–42).”

Comment 2: *Despite being provided with the cross name, selection history and pedigree in the supplementary table, It may have been good to explain a bit more in general terms what the different classes of exotic material were: e.g. do the synthetics all contain a complete D-genome from a diploid wild relative? Do the alien introgressions have single chromosomal segments introduced? Are the elites all contemporary wheat cultivars? I suggest inserting a sentence/section to provide an overview of the genetic composition of the exotics.*

We have added additional details on the different classes of exotic material as well as details on how the Elite lines were chosen (lines 363-373 in revised manuscript):

“These exotic lines include primary synthetic derivative lines, containing between 0.5% and 43% donor material (25); Mexican and other origin landraces derivatives; and Elite lines containing an introgressed segment of *Th. ponticum* on chr7D and/or *S. cereale* on chr1B (25). The set of Elite lines contain 11 CIMMYT varieties released from 1966 until 2007 and additional lines derived from the systematic screening under yield potential and heat stressed field conditions of CIMMYT breeding and pre-breeding material. This allowed the identification of elite genotypes with favourable expression of traits of interest such as high biomass/RUE at different growth stages including final above ground biomass under both yield potential and heat stressed conditions. In general, pre-breeding material is derived from crosses where one of the parents was selected for expressing low canopy temperature and/or high yield or biomass under heat stressed environments.”

Comment 3: *I had some difficulty understanding Figure 4a and the issues associated with genotyping the lines by TENSEQ. (e.g. Why do all the core introgressed regions – esp the key MTA - have ‘expected coverage’?). I appreciate I may simply not understand whats been done – but if that’s the case I have to assume that many others will be in the same boat. Of course, It would have been a lot easier had the original parents been recorded, maintained and genotyped alongside the experimental lines and not having these could be considered a significant flaw. I had to go back and read the earlier papers to try to grasp what had been done and why. The point is I wouldn’t expect a reader to have to do this - so despite the genotypic information having been described previously - I would recommend the authors take some time to carefully craft a summary description of what was done – and why it had needed to be done that way - in terms that are understandable by the first time reader.*

By expected coverage, we mean that the mapping coverage is in line with the median mapping coverage of the panel and thus not sufficiently different in synteny. Regions within an introgressed block that are highly syntenic and similar between wheat and *Ae. tauschii* won't

show high levels of mapping issues and so will have coverage deviation values of around 1. Several small regions of reduced mapping coverage across an introgressed region leads to an overall signal of mapping coverage dropout indicative of a block of divergent DNA.

The reason we specifically mention that a portion of the core region has expected coverage is because this matches up to what we see in terms of synteny between *Ae. tauschii* and wheat, as shown in the synteny dot plot in fig 4b. This plot shows that synteny varies across the region and the regions that have expected or deviating coverage will mirror this if the inserted material is indeed *tauschii*, which it indeed does.

We appreciate that this could be explained better so we have added the following details (lines 191-196 in revised manuscript):

“Importantly, an introgressed region, particularly from a primary genome species such as *Ae. tauschii*, will likely contain some windows without reduced mapping coverage due to high similarity between wheat and the introgressed chromosome segment at those positions. However, these drops in individual windows over a block, in conjunction with species-specific SNPs, allows us to accurately locate the size and position of introgressed segments.”

Sequencing all the original parents would have been unfeasible and beyond the scope of what analyses were intended at the beginning. It also wouldn't have particularly added anything to the identification of the *Ae. tauschii* introgression in particular, beyond perhaps identifying the exact genes that were introgressed if we had a genome assembly of that particular accession.

To make the methods understandable without having to read the previous papers, we have fleshed out various aspects of the methods sections 'DNA extraction, capture enrichment and genotyping', and 'Genome-Wide Association Study (GWAS)' to make them understandable on their own (lines 453-490 in revised manuscript):

“DNA extraction, capture enrichment and genotyping

All genotyping data was taken from (25). Flag leaf material from 10 plants per line was collected from field grown plots post anthesis and pooled prior to extraction with a CTAB-based protocol. DNA was extracted using a standard Qiagen DNEasy extraction preparation and quality and quantity assessed using a NanoDrop 2000 (ThermoFisher Scientific) and the Quant-iT™ assay kit (Life Technologies). From this DNA, dual indexed TruSeq libraries with an average insert size of 450bp were produced for each line and enriched using a custom MyBaits 12Mbp (100,000 120bp RNA probes) enrichment capture synthesised by Arbor Bioscience and using 8x pre-capture multiplexing. 90,000 of these probes were designed using an island strategy to

target regions across the whole genome. A subgenome-collapsed reference was used to design these probe sequences to enable homoeologous regions to be targeted with a single probe, thus expanding the design space. The final 10,000 probes were designed for selected genes, targeting both the gene body and 2Kbp upstream. Post enrichment libraries were sequenced using an S4 flowcell on an Illumina NovaSeq6000 producing 150bp paired end reads.

Sequencing reads were trimmed and low-quality reads removed. These reads were mapped to the Chinese Spring RefSeq v1.0 wheat reference genome (27) using BWA mem v0.7.13 (53). Samtools v1.4 (54) was used to remove unmapped reads, supplementary alignments, improperly paired reads, and reads that didn't map uniquely ($q < 10$). PCR duplicates were removed using Picard's MarkDuplicates (55). SNPs were called using samtools mpileup and bcftools call (56) with parameter -m. SNPs were filtered using GATK (55) to remove SNPs that were heterozygous, had a minimum quality < 30 or a minimum depth < 5 . A locus was designated as homozygous reference if no alternative allele was found but 5 or more reads were mapped at that position. To create a set of shared SNPs for use in GWAS, SNPs for all lines were combined and loci with more than 10% missing data and a minor allele frequency (MAF) below 5% were removed. The remaining SNP loci were subjected to imputation using Beagle 5.0 (57)

Genome-Wide Association Study (GWAS)

STRUCTURE v2.3.4 (58) was used to genetically infer the population structure of the panel and produce a population structure matrix. An admixture model was selected and run using 30,000 burn-in iterations and 50,000 Markov Chain Monte Carlo (MCMC) model repetitions for assumed subpopulations of 2-10 for 10 randomly selected, seeded iterations for each assumed subpopulation. The delta k method from (59) was applied to all 10 replicates to identify the most likely number of definable subpopulations. This was implemented using the STRUCTURE HARVESTER Python script (60). Finally, CLUMPP v1.1.2 (61) was used with 10 independent STRUCTURE replicates to produce a consensus Q matrix for each assumed subpopulation number. GWAS analysis was conducted using the MLM model implemented in GAPIT (62). Principal component analysis eigenvectors 1-10 or membership coefficient matrices for 3-8 assumed subpopulations deduced above by STRUCTURE (64) were used as covariates in the model to mitigate the effects of hidden familial relatedness. The EMMA method (63) was implemented in GAPIT to create a positive semidefinite kinship matrix required by the MLM model. Each MTA flanking interval was deduced by identifying the SNP position furthest upstream and downstream from the highest associated SNP that was above the $-\log P$ threshold of 5."

Comment 4: It wasn't clear to me if the HiBAP lines that were declared as heterozygous on 6D were actually heterozygous. Could they have been mixed samples due to sampling leaves from

10 different plants? Can you distinguish between a heterozygote and a paralogous duplication? Could they be segmental additions located elsewhere in the genome? Why would they be maintained as hets (chance?). Was segregation analysis conducted to test for true heterozygosity? A little discussion around the hets may be valuable.

This is a very good point and something we failed to consider. Although many of the HiBAP lines are expected to be inbred to at least F9 or F10, some lines may encompass more heterogeneity than others. Therefore, it is possible that a pooled sample of 10 plants could give false heterozygosity calls caused by a line being a mix of individual plants segregating for a homozygous introgression. We think the mixed samples are the more likely cause of any false heterozygous calls instead of duplications or segmental additions that are polymorphic between individuals of the same genotype.

We have added a few lines on this point in the discussion section (lines 306-310 in revised manuscript):

“We identified several individuals that appear to be heterozygous for the introgression and for the favourable allele at 6D. As sequencing was conducted on pooled samples of 10 individuals per line, lines that appear heterozygous might instead be segregating for presence/absence of a homozygous introgression. However, we are unable to determine whether these lines are truly heterozygous without sequencing individual plants.”

Comment 5: *I appreciate the desire to try to link regions of genetic association with candidate genes but personally I would play this down a bit – especially for the 1B and 2B associations – as it is all very speculative (was this 2B association the same as detected in Joynton et al 2021 for total chlorophyll content). The good news is that the 6B region is apparently recombinationally active so narrowing the interval genetically should be possible leading to delineation of strong and plausible candidates.*

This is a fair comment and we agree that candidate genes are by their very nature speculative. We feel that an exploration of candidate genes underneath the 6D MTA contributed by the *Ae. tauschii* introgression is valuable to demonstrate the use of a non-reference genome for compiling candidate gene lists and demonstrates that the candidate gene previously considered isn't within the core introgressed region. Among the candidates was a novel *tauschii* gene which we feel is interesting as it shows the difference in gene content between species and the importance of not relying on genes present in the reference genome of the species under study. To reduce the emphasis on candidate genes, we have removed a lot of detail on speculative functional links between candidate genes and heat tolerance from the results section. This information has been put in supplementary note 1 to provide a record of why the genes were

chosen. The discussion paragraph on candidate genes was also edited to be shorter and acknowledge the speculative nature of candidate genes (lines 336-349 in revised manuscript):

“Following the identification of the core introgressed region, we extracted the syntenic region from five *Ae. tauschii* chromosome-level assemblies and used these, as opposed to the wheat reference genome, as our source for putative candidate genes underlying the 6D MTA. Extensive literature searches on the introgressed *Ae. tauschii* genes or the wheat genes within the interval (for the 1B and 2B MTAs) uncovered several candidate genes for further dissection. Candidate genes are by their nature speculative but may provide a starting point for follow-up studies aiming to map the causal genes. As the 6D *Ae. tauschii* segment appears to be actively recombining, it should be possible to precisely dissect this region and map the causal gene. Our proposed candidate genes for the 6D MTA differ from the gene proposed by Singh et al. (13). By using the *Ae. tauschii* genomes rather than relying solely on the wheat reference genome, we have demonstrated that the isoflavone reductase gene Singh et al. proposed is not present in the core introgressed region. This difference and the novel introgressed candidate gene identified highlight the importance of considering non-reference genomes downstream of a GWAS, particularly when divergent material has been introduced, as the variation underlying the trait of interest might be absent from the reference genome. “

The 2B association is not the same as detected in Joynson et al., 2021. We have added a sentence (lines 304-305 in revised manuscript) to state the lack of any overlap between these and previously identified associations, as per minor comment 1 below:

“These MTAs do not overlap with MTAs previously identified for HiBAP I for photosynthetic efficiency (25) or biomass traits (21).”

Minor comments:

Minor comment 1: *Linking to the results (MTAs) of work done previously on the same population could be more apparent.*

There were no overlaps between these MTAs and those we previously identified. We have added the following sentence to add this information (lines 305-306 in revised manuscript):

“These MTAs don’t overlap with MTAs previously identified for HiBAP I for photosynthetic efficiency (25) or biomass traits (21).”

Minor comment 2: *Figure 1: Need a labels in the figure legend*

We believe this is referring to missing expansions of some of the trait names in figure 1b. We have added the full name for each of these traits in the figure legend.

Minor comment 3: *P7 - MTA – expand to Marker Trait Association on first use*

This is already expanded to Marker Trait Association on the first use on page 3 (page 3, line 83 of revised manuscript).

Minor comment 4: *P8 – explain how you define minor and major alleles on first use – e.g. were the minor alleles always from the exotic lines? Were the 1B and 2B positive alleles from the exotic lines (I missed this)*

We thank reviewer 2 for this suggestion. We agree that it is slightly confusing. The minor allele is the favourable allele at all three positions. The 1B and 2B favourable alleles were almost always found in exotic-derived lines, which is noted on lines 183-185 in revised manuscript:

“The favourable allele at each of these MTAs is predominantly found in exotic-derived lines with 50/55 (1B), 44/45 (2B) and 33/33 (6D) lines with the minor allele classified as exotic-derived (**Data S1**).”

To address the confusion between minor allele and favourable allele, we have changed minor allele to favourable allele.

Minor comment 5: *L178 word missing (introgression??)*

We have amended the sentence as follows (lines 202-205 in revised manuscript):

“By comparing the overlapping segments between lines, we found a 1.80Mbp core introgressed region between 5.05Mbp and 6.85Mbp that is present in all lines with the T/T or A/T genotype at 6D-6276646 and absent in all the lines with the A/A genotype (**Fig. 4a**).”

Minor comment 6: *please clarify how does the 1.49Mbp Ae. Tauschii segment relate to the 5.05 Mbp region that you previously defined by using the borders of overlapping introgressions*

We appreciate this comment because we now realise there was a typo that may have led to confusion, in addition to general lack of clarity. The typo has been addressed in response to minor comment 5 above. In the original manuscript we wrote that we found a 5.05Mbp and 6.85Mbp core introgressed region instead of a 1.80Mbp region between 5.05Mbp and 6.85Mbp.

The 1.49Mbp segment is the same region as 1.80Mbp region in the wheat Chinese Spring genome but as it appears in *Ae. tauschii*, in which it is slightly shorter due to structural change in this region between wheat and *Ae. tauschii*.

In addition to the change made for minor comment 5, we have made some changes within the same paragraph to clarify how this 1.49Mbp region relates to the 1.80Mbp region of wheat (lines 207-211 in revised manuscript):

“Using chromosome and protein alignments, we anchored this 1.80Mbp core region from the wheat RefSeq v1.0 genome to the *Ae. tauschii* reference genome, Aet v4.0 (26), and extracted the syntenic 1.49Mbp region between 4.63Mbp and 6.12Mbp. This represents the probable introgressed content of the core introgressed region and likely contains the gene(s) responsible for the MTA (**Fig. 4b, 4c**).”

Minor comment 7: *L189 there is a T missing after Chr6D – 6276646*

Added T after favourable to read favourable T allele (line 213 in revised manuscript):

“Visualising *Ae. tauschii* introgressions across the first 50Mbp of chr6D in six HiBAP I lines, four containing the favourable T allele at chr6D-6276646 (HiBAP 57, 29, 48, and 65) and two containing the unfavourable A allele at chr6D-6276646 (HiBAP 92 and 103).”

Minor comment 8: *L205 – delete ‘from’*

Deleted. Now reads (lines 231-232 in revised manuscript):

“Between accessions, this region varies between 1.49Mbp and 1.82Mbp in length and contains between 26 and 33 genes (**Table S9**).”

Minor comment 9: *L211 – delete 11*

Deleted (lines 244-246 in revised manuscript):

“Within the 2B interval, we identified DEHYDRATION-RESPONSIVE ELEMENT-BINDING PROTEIN 1A (DREB1A) and STEROL GLUCOSYLTRANSFERASE (SGT) as promising candidate genes.”

Minor comment 10: *L245 – first mention of de novo SNP discovery need to introduce – even if repeating what has been said in other publications*

We agree that this could use some introduction. The start of this discussion section now reads as follows (lines 290-292 in revised manuscript):

“*De novo* SNP discovery is the process of generating SNP markers from high-throughput next-generation sequencing as opposed to using lower density genotyping arrays. The value of this approach in breeding efforts is starting to be more widely recognized.”

Minor comment 11: *L 390-391 – replace ‘confirm details’*

Removed. Now reads as follows (line 430 in revised manuscript):

“NDVI and CT orthomosaics were obtained from the aerial images using the software Pix4D.”

Minor comment 12: *L 480 insert AND*

We have corrected this by removing the word “analysed” rather than and. It now reads as follows (lines 537-539 in revised manuscript):

“Protein domains were identified using HMMER hmmscan (73) via ebi using Pfam, TIGRFAM, Gene3D, Superfamily, PIRSF, and TreeFam databases.”

Minor comment 13: *Finally: Why were the genes in the 6D region not submitted to Knetminer*

As the 6D region came from *Ae. tauschii*, it included genes not present in wheat and thus couldn't be used as input to Knetminer so the exploration of candidate genes was done separately to the others using extensive literature searches of the genes and their orthologues from other plant species.

REVIEWERS' COMMENTS:

Reviewer #1 (Remarks to the Author):

It appears to me that the authors have addressed the concerns raised by myself and the other reviewer. Their responses in the rebuttal letter are thoughtful and thorough and therefore I am ready to recommend acceptance of this interesting manuscript.

Reviewer #2 (Remarks to the Author):

The authors have addressed the major concerns raised in my original review. A few minor issues remaining are:

Remaining minor issues/changes

P3: CHANGED TO: Finally, we identify introgressed *Ae. tauschii* introgressions underlying an MTA and employ a novel method downstream of GWAS, using *in silico* mapping to narrow down the interval, explore recombination and identify candidate genes

P9: 'Importantly, an introgressed region, particularly from a primary genome species such as *Ae. tauschii*, will likely contain some windows without reduced mapping coverage due to high similarity between wheat and the introgressed chromosome segment at those positions. However, these drops in individual windows over a block, in conjunction with species-specific SNPs, allows us to accurately locate the size and position of introgressed segments.'

This remains a very clumsy and contradictory couple of sentences and should to be rewritten. The first sentence is OK – the second is where the issue arises (ie the first discusses regions of high similarity, the second regions of low similarity).

P10. Heterozygotes lines 205-207. Should you introduce possible explanations for observed heterozygosity here?? How do the heterozygous families respond to heat treatment in comparison the homozygotes?

P13 L309: 'However, we can't determine whether these lines are truly heterozygous without sequencing individual plants. Couldn't you simply look at segregation of the heterozygous markers in the subsequent generation?'

See previous comment – were the het families phenotypically indistinguishable from the homs in terms of heat tolerance or was there variation within the families (I presume the leaves used for genotyping were taken from the plots used for phenotyping??)? This could tell something about the possible mechanism (e.g. dominant vs. recessive) that may allow further prioritisation of candidate genes/alleles for validation.

Dear reviewer 2,

Thank you very much for your feedback. Below are responses to your remaining two concerns. Our response is in red. In the new revised manuscript, these changes are highlighted with track changes.

P3: CHANGED TO: Finally, we identify introgressed *Ae. tauschii* introgressions underlying an MTA and employ a novel method downstream of GWAS, using in silico mapping to narrow down the interval, explore recombination and identify candidate genes

P9: 'Importantly, an introgressed region, particularly from a primary genome species such as *Ae. tauschii*, will likely contain some windows without reduced mapping coverage due to high similarity between wheat and the introgressed chromosome segment at those positions. However, these drops in individual windows over a block, in conjunction with species-specific SNPs, allows us to accurately locate the size and position of introgressed segments.'

This remains a very clumsy and contradictory couple of sentences and should to be rewritten. The first sentence is OK – the second is where the issue arises (ie the first discusses regions of high similarity, the second regions of low similarity).

We agree this was poorly written. We have changed a little more than the one sentence to try and make it more understandable. It now reads as follows (lines 162-175 in revised manuscript):

“Due to the better performance of exotic-derived lines under heat stress and exotic-derived lines possessing alleles for heat tolerance, we searched for introgressed material overlapping the MTAs. We detected introgressed material in HiBAP I lines by looking for genomic blocks containing windows with SNPs specific to *Ae. tauschii*, *Th. ponticum* or *S. cereale* and reduced mapping coverage, seen as coverage deviation (mapping coverage compared to the median mapping coverage across the panel) significantly below 1, which indicates breaks in synteny between wheat and the introgressed chromosome segment. Using this approach, we identified

introgressed *Ae. tauschii* material at the beginning of 6D in all 33 lines with the T/T genotype and all 7 lines with the A/T genotype at MTA 6D-6276646, where T is the favourable allele. As *Ae. tauschii* is from wheat's primary genome and thus very similar to the D subgenome, not every 1Mbp window is sufficiently lacking in synteny for reads to map poorly and produce significant coverage deviation below 1. This explains why some windows within the introgression have coverage deviation of around 1. However, these windows still have *Ae. tauschii*-specific SNPs and are within a block of 1Mbp windows in which most have significant coverage deviation below 1. Therefore, we can be confident that the introgression includes these windows."

P10. Heterozygotes lines 205-207. Should you introduce possible explanations for observed heterozygosity here?? How do the heterozygous families respond to heat treatment in comparison the homozygotes?

P13 L309: 'However, we can't determine whether these lines are truly heterozygous without sequencing individual plants. Couldn't you simply look at segregation of the heterozygous markers in the subsequent generation?'

See previous comment – were the het families phenotypically indistinguishable from the homozygotes in terms of heat tolerance or was there variation within the families (I presume the leaves used for genotyping were taken from the plots used for phenotyping??)? This could tell something about the possible mechanism (e.g. dominant vs. recessive) that may allow further prioritisation of candidate genes/alleles for validation.

We found that the heterozygous and homozygous lines are not significantly different in yield or canopy temperature under heat stress, suggesting the phenotype may be dominant and adding support that the lines are heterozygous, and it isn't just an artefact of the sample pooling.

Changes in the text have been made in the results and discussion as below:

In the results following the first reporting of the favourable alleles (lines 154-160 in revised manuscript):

“7 lines appear to be heterozygous (A/T) at 6D-6276646. The HiBAP lines are inbred to at least the F9 or F10 generation so, as sequencing data was generated from pooled samples, this observation could be the result of alleles segregating at this locus. However, we observe no significant difference in yield or canopy temperature under heat stress between lines that are heterozygous and lines that are homozygous for this allele (Supplementary Figure 3). This suggests that these lines are indeed heterozygous for the favourable allele and also suggests that the phenotype may be dominantly inherited.”

Supplementary Figure 3. Yield and vegetative canopy temperature under heat stressed conditions for lines with homozygous unfavourable allele (A/A), heterozygous for the favourable allele (A/T) and homozygous for the favourable allele (T/T). Black points indicate individual data points; this was done due to the small sample size of A/T. Significance was computed using a one-way ANOVA test (n=109, 7, and 32 biologically independent lines for A/A, A/T and T/T, respectively). Tukey’s honest significance test was used to calculate adjusted p-values for each pairwise comparison.

in the discussion (lines 295-303 in revised manuscript):

“We identified several individuals that appear to be heterozygous for the introgression and for the favourable allele on 6D. As sequencing was conducted on pooled samples of 10 individuals per line, lines that appear heterozygous might instead be segregating for presence/absence of a homozygous introgression. As the phenotype under heat stress appears to be the same between lines that are homozygous and lines that are heterozygous at this locus, it seems likely that these lines are heterozygous for the introgression and allele. In addition, this also indicates that the heat tolerant phenotype contributed by 6D may be dominantly inherited; however, additional work would be needed to validate this. The zygosity of the allele in these lines can be verified in future work by developing markers and observing how they segregate in subsequent generations.”